

# Evaluation of Environmental Factors in Landslide Prone Areas of Central Taiwan using Spatial Analysis of Landslide Inventory Maps

**Kui-Lin Fu[1], Bor-Shiun Lin[2,*], Kent Thomas[3], Chun-Kai Chen[4] and Hsing-Chuan Ho[5]**

[1]  Ph.D. Candidate, Department of Soil and Water Conservation, National Chung Hsing University

  Address: 145 Xingda Rd., South Dist., Taichung City 402

  TEL: (886-49) 234-7200   FAX:(886-49) 239-4306

  E-mail:fgl@mail.swcb.gov.tw

[2*] Principle Researcher, Disaster Prevention Technology Research Center, Sinotech Engineering Consultants, Taiwan.

  Address: No. 280, Xingu 2nd Rd., Neihu Dist., Taipei City, Taiwan, R.O.C., 11494.

  Corresponding author: TEL: (886-2) 8791-9198 EXT.309  FAX:(886-2) 8791-2198

  E-mail:bosch.lin@sinotech.org.tw

[3] Building Engineer – Infrastructure Services; Roads, Buildings and General Services Authority (BRAGSA).

  Address: Cnr. Lower Bay Street & McCoy Street, P.O Box 1100, Kingstown, St.Vincent and the Grenadines.

  TEL: (784) 430-0451 E-mail: kent.gary.thomas@gmail.com

 [4]Associate Researcher, Disaster Prevention Technology Research Center, Sinotech Engineering Consultants, Taiwan.

  Address: No. 280, Xingu 2nd Rd., Neihu Dist., Taipei City, Taiwan, R.O.C., 11494.

  TEL: (886-2) 8791-9198 EXT.307  E-mail:ckchen@sinotech.org.tw

[5] Associate Researcher, Disaster Prevention Technology Research Center, Sinotech Engineering Consultants, Taiwan.

  Address: No. 280, Xingu 2nd Rd., Neihu Dist., Taipei City, Taiwan, R.O.C., 11494.

  TEL: (886-2) 8791-9198 EXT.314  E-mail:hcho@sinotech.org.tw

**Abstract**

For many years the Shenmu watershed has been heavily impacted by landslides induced by extreme rainfall events, with an even greater impact in recent years due to climate change in addition to the Chi-Chi Earthquake after-effects. This study utilizes remote sensing technology to spatially and temporally interpret landslide processes in the Chushui and Aiyuzih sub-watersheds. An event-based landslide dataset is constructed which consists of 11 historical disaster events including 17 satellite images spanning the past 14 years. The study





explores the contribution of causative environmental factors and other factors, which are
based on the physiographic conditions and geographic locations of the landslides induced, on
landslide potential. These factors are utilized to construct a logical reason-based rule set to
build a framework of procedures for semi-automated image interpretation and artificial image
identification.
Spatial relationships show that landslides are frequently found in areas at 1500m ~2000m of
elevation with slope gradient over 55% W-SE orientation, and within the Nanchuang
Formation adjacent to a 25m buffer zone of a river course in the Chushui and Aizuyih sub-
watersheds. Landslide occurrences are prevalent on both sides of the river course and are the
direct suppliers of sediments to the river, leading to sediment related disasters. Temporally, it
is found that the typhoon-induced landslides can be subdivided into three distinct time
intervals and the event that caused the greatest increase in landslide area can be recognized.
These intervals and their greatest impact events are as follows: (1) Before the 1999 Chi-Chi
earthquake, the 1996 typhoon Herb; (2) From the 1999 Chi-Chi earthquake to 2009 typhoon
Morakot, the 2004 typhoon Mindulle; (3) After this time period, the 2009 typhoon Morakot.
After comparisons were made of the total landslide areas and the new landslide areas before
and after the 1999 Chi-Chi earthquake in the two sub-watersheds, it was found that the
earthquake amplification effect of the quantified magnification was estimated to be at least a
doubling effect. This is an estimate that agrees well with the previous studies. The
methodology of our extensive study can be utilized to improve the dataset accuracy in similar
research, to classify and differentiate the contribution of environmental factors to landslide
occurrences and to build landslide occurrence potential maps for sub-watersheds. These
results are important to decision-makers to improve the reference information basis for
preliminary evaluation of the exposure of elements at risk. This, in turn, is important for
improving early warning systems, rapid response mechanisms, evacuation protocols and long
term mitigation solutions. The results may also influence the recommendations for
remediation of slope areas and construction of preventative engineering solutions in the two
sub-watersheds analyzed using the landslide potential map to prioritize urgencies by
comparing the necessity in one area to the next.
Keywords: Shenmu watershed, environmental factors, landslide potential map.



## 1 Introduction

Since the 1996 typhoon Herb, Shenmu Watershed has experienced multi-temporal events such as the Chi-chi Earthquake, typhoon Toraji, typhoon Mindulle, typhoon Morakot, etc. These increased the landslide area in a watershed scale. Sediment production from bare and forested lands has tremendous impact on the objectives of preservation at downstream. Triggering factors are the primary contributors associated with the event which initiate or induce the slope failure or landslides, such as rainfall or earthquake events. Causative factors are the secondary contributors which increase the prevalence of landslide hazards, such as geology and distance from fault line. In some cases, causative factors may induce slope instability and failure such as rivers undercutting slopes and thus, can be classified as primary factors. This research primarily focuses on the causative factors in the Shenmu area. Long term monitoring of topographic changes was conducted via multistage satellite imagery of landslide areas to analyse landslide evolution and correlation to causative factors. Since the 1980s, many researchers have studied landslide evolution, correlation of causative/triggering factors to landslide zonation, landslide potential and probability using remote sensing technology (van Westen *et al.*, 2008). These studies joined local data with multistage DEM data generated by airborne and satellite-based optical sensors to monitor topographical evolution and landslide activity of high-mountain terrain deformation, to collect causative/triggering factors data and to compare their contribution to landslide events which can be utilized for early warning of hazards and reduction to disaster risk (Carrara, 1992、1999;Montgomery and Dietrich, 1994). Kääb (2002) and Chau *et al.* (2004) estimated the landslides zonation and population at risk according to landslide susceptibility map that quantifies hazard-inducing factors such as elevation, slope gradient, rock characteristics, debris deposits, population distribution, climate and rainfall. Metternicht *et al.* (2005) conducted long-term monitoring of landslide hazards, deployment of landslide database, and studies of hazard-inducing factor correlation in Sweden. Barlow *et al.* (2006) illustrated correlation between landslide and environmental factors by quantifying the characteristics of mass movement and determining based on normalized differential vegetation index (NDVI) and terrain slope gradient. Bai *et al.* (2009) utilized landslide-triggered factors to establish the landslide susceptibility map for the Three Gorges Dam in the Yantze River. Mehrnoosh *et al.* (2009) established a landslide potential map which demonstrates that the primary landslide-





triggered factor is formation lithology and also concluded that soil profile as well as human
development is only secondary.
In Taiwan, Chen and Cheng (1997) used three periods of SPOT satellite images of the
Fungshan river watershed combined with GIS to quickly extract the evolution data from large
hillslope development within the watershed to effectively monitor land use change and
vegetation coverage evolution within the watershed. The National Science and Technology
Center for Disaster Reduction (2004) used satellite images of post typhoon Aere event to
digitize and calculate the incremental landslide area in Shihmen Reservoir watershed to
conduct research on correlation among landslide size, landslide zoning, human activity,
accumulated rainfall, etc.. Taiwan Soil and Water Conservation Bureau (SWCB, 2010[a])
reported that typhoon-induced landslide areas within Shih-men Reservoir watershed are
concentrated between 1,000m to 1,500m, mostly on slopes steeper whose geology mostly
belongs to Taliao Formation.
For effectively evaluating causative factors of landslides, this study compiles all the historical
satellite images of the Shenmu watershed and uses a proposed framework for semi-automated
image interpretation and artificial image identification procedures which systematically
improves the quality of data content and data structure so that it can definitely promote data
integrity and accuracy for improving value and reliability of landslide inventory with pre and
post various triggering events, meteoric, seismic and environmental. Secondly, using spatial
analysis with GIS-based program can explore or extract environmental factors of the collected
spatial data layer and DEM materials to qualify and construct such potential maps. Finally,
these results would help decision-makers to improve the reference information to forecast the
occurrence of future landslide basis for preliminary evaluation of the exposure of elements at
risk. This in turns would be important for improving early warning systems, rapid response
mechanisms, evacuation protocols and long term mitigation solutions.
**2   Hazard History of Study Area**
Shenmu watershed is situated in the southwestern corner of Xinyi Township in Nantou
County. Traffic access mainly depends on the Highway 21 which goes north to Shueilee
Township. Shenmu watershed is highly mountainous with steep slopes. Local topography is
characterized by tall mountains and steep slopes, with elevations ranging from 500 m a.s.l. to





2500 a.s.l. and slopes which are steeper than 28.8° represent 45% of the total regional
extension (72.16 km$^2$) and 47.75% of the slopes are north facing (Ho *et al*., 2011; Lo *et al*.,
2012). The Shenmu watershed is crossed by three primary geologic structures: the northeast-
southwest Heshe Anticline and Tungfu Syncline and the Chen-yo-lan River Fault. These
mountain slopes are covered with dense forests and were built up by the Nanchuang and
Heshe formation (see Fig. 1). These formations consists mainly of hard, dark grey argillite
and grey slate with thinly bedded muddy sandstone, which are prone to severe weathering and
become weak layers in the rock strata. The Shenmu watershed is located in the Heshe river
watershed, an upstream watershed of the Chen-yo-lan river. The Aiyuzih, Housha and
Chushui rivers constitute the Heshe river watershed. Temperature ranges from 5.9℃ ~ 14.4℃,
averaging 10.9℃ annually. The average annual accumulated rainfall for three weather
monitoring stations located at Alishan, Shenmu Village, and Hsinkaoko ranges from 1,950 to
4,980mm.
In 1996, typhoon Herb brought lots of rainfall up to 714 mm over 3 days that causes the first
occurrence of massive landslide. Chushui and Aiyuzih rivers, which are in upstream portion
of the Shenmu watershed, deliver massive amount of sediment material which converged and
flowed toward the mid and downstream of Heshe River. Consequently, Shenmu Elementary
School which was near the converging point was buried by the debris, and the checkpoint and
Shenmu Bridge were both destroyed. The total out-flowing debris was 450,000 m$^3$. Thereafter,
in May 1998, the Hosa River Bridge in front of Shenmu Elementary School was destroyed by
the debris flow from Chushui River. 155 people from 39 households were stranded, causing
large scale panic and forcing nationwide attention to debris flow hazards. Similarly, in May
1998, the plum rain season (i.e. in the East Asia region, it coincides with the season of plums
ripening) carried massive rainfall which caused the Hosa River Bridge to be destroyed by
debris flow so that over 100 residents were isolated. The 2001 typhoon Toraji brought
accumulated rainfall of 517mm, and the rehabilitated Hosa River Bridge was once again
destroyed. Highway 21 access was disrupted and there was extensive damage to housing and
farms. Between May and June 2004, Shenmu Watershed had four debris flow events due to a
series of continuous rainfall events. This caused the groundsill structures deployed by the
Taiwan Forestry Bureau at upstream Aiyuzih River to be buried, causing severe sediment
material on the two sides of river flank. In July of the same year during typhoon Mindulle



whose total accumulated rainfall was up to 1,254 mm, there was large-scale landsliding at
upstream Aiyuzih River and large amount of sediment material deposited in the river course.
In June 2006, two days of non-stop rainfall brought 1,332 mm of accumulated rainfall to
Shenmu watershed. Chushui River once again had debris flows and large-scale landsliding
that damaged cultivated farmland. On August 6th, 2009, typhoon Morakot gradually passed
through Taiwan. Its intensity steadily increased and continued to move to the west. Typhoon
Morakot had a complete structure and moved slowly with accumulated rainfall duration of six
days (2009/08/05~08/10). Also, on the night of August 8th, the system's accumulated rainfall
reached 900 mm which exceeded the 200-year rainfall return period. This caused the river to
surge, severe scouring on the two flanks, several collapses on the roads and massive sediment
washed down destroying the downstream Aiyuzih Bridge. The disaster points of this event are
investigated as shown in Figure 2. Consequently, Shenmu Watershed almost was like an
isolated island without communications and local residents had a rescue and less resource
supply for livelihood. In a view of above the disaster history, Shenmu area is generally
affected by debris flows during the typhoon and flood seasons and has the highest debris flow
frequency throughout Taiwan, especially within Chushui and Aiyuzih subwatersheds.
This study aims to analyze the rainfall event-induced and environment factors correlated with
landslide evolution of Chushui and Aiyuzih subwatersheds to serve as a reference for disaster
prevention and management.
**3   Spatial Data and Methodology**
Rapid advances of computer technology have promoted similar improvements in remote
sensing (RS) techniques and geographic information systems (GIS). Multi-stage RS images
can be effectively applied to practical issues such as landuse planning and mapping, detection
of geomorphological change and wide-area disaster monitoring. Also, in terms of efficiency
and cost, remote sensing is superior to traditional methods especially for collecting and
processing data over large areas. Hazard zonation is the fundamental basis of all modern
disaster management and preparedness strategies and provides a basic knowledge of potential
danger of specific events in a given area. Remote sensing has proven to be very effective in
performing rapid, emergency data collection during post disaster recovery periods. As a result,
countries around the world are increasingly using remote sensing to perform prompt large-
scale post-event natural disaster surveys. Van Westen (2000) has pointed the interpretation of



post-event residual characteristics, such as landslide scars, interpreted and confirmed using
RS and field investigation studies then consolidated into GIS databases create the foundation
of hazard zonation. Other authors have further suggested improvements to hazard zonation by
including various other datasets, such as borehole data, geological data, knowledge derived
from local communities and past events (Fell *et al.*, 2008). These hazard maps should provide
local authorities, communities, decision-makers and stake-holders (such as private companies
and NGOs) with comprehensible information which can be utilized for planning. Van Westen
et al. (2008) have also suggested that longterm RS data, GIS data and satellite imagery
projects can further enhance hazard mapping by providing spatial and temporal relationships
which map the variability and evolution of geomorphological, environmental and agricultural,
river, man-made development and settlement datasets. The issue faced by many countries is
the lack of longterm data, cost of longterm projects and the seeming lack of short term
benefits. These GIS-based information and datasets when adequately utilized can be
organized into easily accessible disaster management systems to improve disaster response,
create early warning/detection systems, utilized for probalistic research and subsequent long
term data analysis purposes, and finally to further enhance the disaster mitigation and
prevention stages of the disaster management cycle.
In this study, the object-oriented Trimble®  eCognition Developer has been used to handle
image analysis tasks, to process a variety of image sources, to provide automatic or semi-
automatic processing and analysis. It is also used in mapping landslide studies after a given
landslide event to develop rule sets for the analysis of remote sensing data (Borghuis *et al.*,
2007; Joyce *et al.*, 2008; Lu *et al.*, 2011). eCognition Developer enables non-technical users
to configure, calibrate and execute image analysis workflows (Trimble, 2010). For landslide
detection, eCognition Developer has specific characteristic for classifiying polygon object
instead of grid-based information. It can execute image segmentation quickly and easily in
different scale under supervised classification and using fuzzy logic classification algorithms
to improve overall accuracy of classified results for a specific event.
In term of spatial and statistical analysis, the landslide inventory gives insight into the
location of landslide phenomena in a specific study area, displaying information on landslide
activity (van Western *et al.*, 2008), and therefore should require multi-temporal landslide
information for specific disaster-prone regions. The occurrence of landslide in a watershed





scale area is directly related to the rainfall distribution, accumulation, duration, hourly
intensity, and its patterns, but is also affected by environmental conditions. Many studies have
focused on the spatial analysis of landslides affected by environmental factors including
morphology, geology, hydrology, geomorphology and human activities (Cruden and Varnes,
1996; Aleotti and Chowdhury, 1999; Ayalew and Yamagishi, 2005). Especially, rainfall is the
primary factor triggering landslides in Taiwan, and can also be used as a key factor in
predicting where and when landslides will occur (Chen et al., 2014). Several studies  show
rainfall event characteristics, such as rainfall duration, rainfall intensity, accumulated and
antecedent rainfall could be quantified as the threshold value for landslide occurrence (Caine,
1980; Crozier, 1986; Jakob and Weatherly, 2003; Chen *et al*., 2014). Chen et al. (2014) have
observed that the 1999 Chi-Chi earthquake and the accumulated rainfall of subsequent
typhoons and heavy rainfall events substantially affected the distribution and severity of
landslides in the Shenmu watershed.
In view of the above, the following sections would exhaustively express our spatial data,
utilized methodology and proposed procedure to evaluate environmental factors in landslide
prone areas of Central Taiwan using spatial analysis of landslide inventory maps.

### 3.1   Landslide Inventory Maps

To build a reliable hazard map to predict landslide-prone areas in Shenmu Watershed, this
study required a landslide inventory that is as complete as possible in both space and time
(Glade, 2001; Malamud *et al*., 2004; van Westen et al., 2008). This study establishes the
event-based landslide inventory using multi-stage satellite imagery interpretation for Chushui
and Aiyuzih subwatersheds in the Shenmu area. Image interpretation defines the
physiographic and geographic nature of area as environmental parameters. These
environmental parameters are joined with the rainfall triggering factors to correlate landslide
proneness. A collection of landslide inventory maps of extreme meteorological events and one
extreme earthquake event (Dadson *et al*., 2003; Lin *et al*., 2008) affecting the Shenmu area is
built containing 11 events from 1996 to 2009 including 1996 typhoon Herb, 1999/5/28 heavy
rainfall, 1999 Chi-Chi earthquake, 2001 typhoon Toraji, 2004 typhoon Mindulle, 2008
typhoon Kalmaegi, typhoon Fung-wong, typhoon Sinlaku, typhoon Jangmi, 2009 typhoon
Morakot and typhoon Parma. (see Fig. 3 and Table 1).

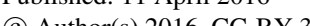



Landslide inventory maps are the key component to correlate environmental parameters with
triggered factors and changes in these characteristics after major triggering events (Coe *et al*.,
2004). Using satellite remote sensing data for the identification and mapping of small-scale
landslide areas has been improved substantially over the last decade. The selection of satellite
imagery sources is an important factor affecting the reliability of identification, satellite
imagery for later events has higher spatial resolution due to more advanced technology of
satellite data acquisition system. SPOT-series satellite has been widely utilized where
landslides without vegetation can be differentiated spectrally from their surrounding areas
(Yamaguchi *et al*., 2003; Nichol and Wong, 2005; Borghuis *et al*., 2007). Accordingly, for
pre-2004 imagery this study adopted SPOT1-SPOT4 satellite imagery whose resolution is
6.25m or 10m and for post-2004 imagery this study has adopted SPOT5 satellite or
FORMOSAT II satellite imagery whose resolution is 2.5m and 2.0m for completion of 1996-
2009 landslide inventory maps (see Fig.4). Besides, all of the satellite imagery data qualities
are required to be low or no-cloud cover, and the capture angle of satellite data acquisition
system is less than 15 degree (SWCB, 2010[b]). Table 2 lists the adopted image data source for
each event. This study, using GIS-based layers of the landslides inventory maps caused by
extreme events and of different physiographic characteristics of the study area, discusses the
long term changes in the Shenmu watershed and explores the correlation between
environmental parameters and landslide area.
**3.2  Image Interpretation and Identification Procedure**
Numerous studies have elucidated the effects of subjective judgment errors made by landslide
researchers using semi-automated methods (Martha *et al*., 2010). In modern landslide
research, the most common approach to landslide inventory mapping is using remote sensing
technology combined with GIS-based programs. This study adopted object-oriented semi-
automated image interpretation to quickly extract temporal changes from pre- and post-
disaster satellite imagery in areas which have been affected by landslides. Then, all of the
extracted areas were filtered out and error-checking was carried out through elaborate
artificial quality control for completing data standardization and promote data accuracy.



### 3.2.1 Semi-automated Image Interpretation

The suggested object-oriented semi-automated image interpretation procedure (see Fig.5) in this study will be organised into four steps described as follows:

a. Import satellite images and reference layer:Before importing the whole inventory of satellite images into this program, the priority action is required to define the local characteristics of landslide area within satellite images depending on some GIS-based reference layers such as road, landuse and DEM-derived slope or aspect for effectively detecting object likely landslid-shape polygons.

b. Optimize object scale size:The imported satellite images are separated regularly by multi-resolution segmentation method, which for a given number of image objects, minimizes the average heterogeneity and maximizes their respective homogeneity into different object size from small to large scale. If the scale size is too small, large amount of objects are separated and too fractured. The same thing would be composed of multiple objects and that would cause subsequent increase in processing time. Conversely, if the scale size is too large, the separated objects are too few that can not divided to be the proper objects for classification which means landslide are hardly detected. Consequently, one shoud set up minimum map unit be greater than 9 to 12 pixels (Desclée *et al*., 2006) to optimize object scale size based on the best resolution of SPOT-series satellite images in practical application.

c. Construct the appropriate classified rule set for landslide detection:when completeing object scale size optimization, the appropriate classified rules for landslide detection need to be systematically constructed. Users should establish feature categories of classification and hierarchy of the tree structure, such as landslide, buildings, road, rice fields, vegetation, river systems, cloud, etc.. Then, it would help to detect the above features based on its logical rule set developing on its own characteristics. In this study, the adopted SPOT-series satellite images has spectral bands, with simultaneous panchromatic and multispectral acquisitions, whose R, G and NIR spectral bands refer to Normalized Difference Vegetation Index (NDVI) for distinguishing non-vegetated area in satellite imagery (Lambin and Strahlers, 1994; Tsai *et al*., 2010). If NDVI value is less than 0.05, there is a high probability that the detected land cover/objects are landslides (see Fig.5). In



addition, the DEM-derived slope for a given area is used to automatically classify and
delineate non-landslide polygons, where slope gradient is less than 5%, and likely-
landslide polygon. Expert judgement is designated artificially as a logical ruleset to
increase the precision of detection of landslide distribution in a temporal and spatial scale.
d. Export classified polygon objects:Classified landslide polygon objects are exported into
GIS-based program as in SHP layer format which include attached information such as
classification attributes, area, aspect ratio, etc., for further artificial checkup, identification
and error elimination.

### 3.2.2 Artificial Image Identification

Artificial image identification procedure refers to interactively adjusting the classified
landslide polygon objects based on visual interpretation with professsional experience,
historical information, aerial photos, land use/land cover and landslide inventory dataset under
the GIS-based program (Tsai *et al*., 2010) such as ArcGIS and Mapinfo. The purpose of this
procedure aims to ensure or promote the overall accuracy of classified reuslts for a specific
event which identify the boundaries of landslide polygons if corresponding with land use and
land cover of current geomorpological environment. Similar to the previous section, the
artificial image identification procedure (see Fig.5) will conform to the following two steps as
described below:
a.  Initial image identification:
This step depends on professional staff with associate geomorphology knowledge and
long-term experience in visual landslide interpretation. Often, some parts of satellite
images may have ambiguities along the boundaries of landslides, especially near existing
stream channels, clouds or shadows in the images that would cause omission of landslide
polygon objects (Tsai *et al*., 2010). For solving this problem, the staff uses aerial photos
and pre-event or post-event images as reference layers to identify if all of landslide
polygon objects have been delineated to minimize the probability of omission.
b.  Advanced image identification:



After intial image identification, some objects belong to artificial agricultural lands or
man-made croplands including bamboo field, tea or vegetable garden. In satellite imagery,
Agricultural croplands and crops which have been cleared (leaving exposed areas) have a
hue that is close to the landslide polygon object and this can cause judgment confusion in
the intial image identification. Therefore, this step utilizes landslide inventory dataset or
land use/land cover map to eliminate the identified landslide polygon objects and
diffrentiate the agricultural croplands for final quality control to reduce the error from the
semi-automated image. Finally, the identified landslide polygon objects is doubly
examined thoroughly to establish a complete landslide inventory map for correlation
analysis with multi-stage landslide inventory maps.
**3.3   Spatial and Statistical Analysis**
The selection of specific environmental factors, which is sometimes referred to as causative
factors, in a given region will strongly dominate the probability for forecasting the occurrence
of future landslide. In this study, using spatial analysis with GIS-based program can explore
or extract environmental factors of the collected spatial data layer and DEM materials (such
as elevation, slope, aspect, geology and roads, etc.) (see Fig.6) overlapping the established
complete landslide inventory map for specific each event. Among these factors of
geomorphological information, a collected high-resoultion DEM was used to calculate/
generate elevation, slope and aspect maps under GIS analysis tools by an interpolation
method. Finally, the above environmental factors were analyzed to identify mutual
correlations with the landslides for a given event and it also can be used to account for the
main cause of landslide and illustrate disater-prone zoning map for effective watershed
management, planning of disaster prevention works and reducing risk of landslide hazards
during the flood season.
To effectively appreciate rainfall-triggered events that cause landslide occurrence affecting
the evolution of landforms and severe topological changes in Chushui and Aiyuzih
subwatersheds within study area, this study compiles satellite images from 12 historical
hazard events from the 1996 Herb to 2009 Parma (see Table 1 and Table 2). According to the
suggested image interpretation and identification procedure, the landslide inventory of both
pre-event and post-event has been detailed mapped into a reliable spatial database. Afterward,





this study estimated possible correlation between landslides and the above mentioned
environmental parameters and connecting landform evolution related to disasters by
performing GIS spatial and statistical analysis to illustrate a series of frequency
distribution/histogram or statistical results for each event. The detailed spatial and statistical
analysis process in this study are illustrated in Fig.6 and the adopted categories of
environmental factors are clearly lised in Table 3.
**3.3.1 Landslide analysis with enviromental facator**
Event-based landslide evolution study compiles the pre- and post-event total landslide areas
and new landslide areas which affected the study area and the corresponding landslide ratio
(*LR*) and new landslide ratio (*NLR*) for a given watershed scale as determined by the landslide
inventory database (Malamud *et al*., 2004; Lin *et al*., 2008; Chang *et al.*, 2014). Pre-event
total landslide areas demonstrate the pre-event stability conditions and represents zones which
further landslide evolution may occur or zones which suffer from wide-scale stability issues.
Post-event total landslide areas demonstrate post-event stability conditions of the study area
and represents both the historical landslides and those landslides formed after or during the
event. New landslide  areas represents the difference of these two and denotes the landslides
directly caused by the event. Some researchers (Chen and Wu, 2006; Chang *et al*., 2007; Lin
*et al*., 2008) used the landslide ratio and new landslide ratio after an event to analyze landslide
condition. The landslide ratio is defined as the ratio of the total landslide area to the given
watershed area. Similarly, the new landslide ratio as the ratio of new landslide area to the total
landslide area. For evaluating environmental factors contribution to landslide evolution, the
above related terminology would be defined accordingly and the schematic layout and its
corresponding explanations are also shown in Fig. 7. The two mentioned ratio equations can
be expressed in percentage respectively as follow:
$$LR(\%) = \frac{TLA}{WSA} \qquad (1)$$
Where $TLA$ is a total landslide area after one specific event including the old landslide area
and new landslide area; the old landslide area is referred to the landslide already exists before
that event; new landslide area is referred to be appeared only after that event; $WSA$ is a given
watershed area.


$$NLR(\%) = \frac{ILA}{WSA}$$   (2)
Where $ILA$ is a toal new landslide area after the specific event.
On the other hand, it is an alternative purpose for rigorously understanding the impact of
environmental factors (see Table 3) on historical rainfall event-triggered landslide inventory
maps, and also evaluate the relationship between the toal landslide area and each single type
or classifications of the same condition. So, this study uses conditional analysis mehtod
(Carrara *et al.*, 1995; Clerici *et al.*, 2006) under GIS-based program incorporate with spatial
and statistical analysis for studying on the landslide contribution from a number of causative
fators existed simultaneously. Clerici *et al.* (2006) studied how factors can be directly or
indirectly related to landsliding by adopting a method of representing factors as a number of
data layers in overlayed order to obtain all the possible combinations of the various classes of
the different factors. The concept of unique condition units (UCU), unique condition subareas
and unique condition classes have been used by different researchers to represent terrain units
compiling the combinations of environemntal factors (Bonham-Carter, 1994; Chung *et al*.
1995). Landslide contribution of environment factors are then obtained in each UCU. A
general assumption used in landslide research is that the conditions which have led to
landslide occurence in the fact are the likely conditions which will lead to future instability
and landsliding. The computed landslide contribution represents causative factors occupied
area entailing with landslide in percentage. Formally, the landslide contribuion ( $LC$ ) is
redefined as (Carrara *et al.*, 1995; Clerici *et al.*, 2006):
$$LC(\%) = \frac{TLA \cap UCUA}{TLA}$$   (3)
Where $UCUA$ means the total area of unique condition unit (UCU).
The study uses the above conceptual equation to assess landslide contribuion of each UCU to
the corresponding environmental condition. The landslide contribution of UCU represents the
degree of influence of any causative factors. Among them, if the contribution extent is high,
this means in the classification, it is a main causative factor which would help verifying where
the main disaster-prone area regarding the assessment of landslide presence, the choice of the
factors to use in the analysis and the evaluation of the reliability of the resulting zonation. All
of the detailed results will be discussed in the following sections.





### 3.3.2 Landslide analysis with rainfall events

Researchers suggest using aerial photo and satellite images to interpret the landslide area to reduce investigation cost. In terms of methodology, there are empirical formula, expert judgment, mechanism approach, and statistical approach (Uchihugi, 1971;Montgomery and Dietrich, 1994;Aleotti, 2004). Among them, empirical formula is most widely used. Uchihugi (1971) used rainfall data from different regions to obtain the formula for accumulated rainfall and landslide ratio. SWCB (2010[a,b]) used Uchihugi's empirical model to quickly obtain disaster sediment production in the watershed. This was compared with the observed data by airborne LiDAR, and results from watersheds with higher rainfall station density were a closer match.

Uchihugi (1971) targeted new landslides in typhoon events for analysis. He discovered that the greater the accumulated rainfall, the larger and more numerous the landslides. Uchihugi adopted the analytical concept and used rainfall data from different regions. The relationship between accumulated rainfalls and landslide area ratio was obtained to simplify and speed up the estimation of new landslide areas. However, when the rainfall parameters of Uchihugi empirical model reach the critical rainfall, the new landslide in the watershed becomes zero. This does not fit with the physical reality. The study extended from Uchihugi concept on the basis that when the watershed accumulated rainfall equals critical rainfall, landslide should happen. Therefore, it included a parameter for initial new landslide ratio, which is the state of landslide when the watershed is under critical rainfall. According to Eq. (2), Uchihugi formula was modified as below:

$$NLR(\%) = \frac{ILA}{WSA} \approx C + K \times 10^{-6} (R_A - r)^2 \quad RI \geq r \qquad (4)$$

Where K is the coefficient and $R_A$ is the accumulated rainfall of an event (mm). r is the critical rainfall for landslide initiation (mm), and C is initial increment landslide ratio when the critical rainfall triggers the landslide.

Due to lack of actual critical rainfall (r) for landslide in this study area, the critical rainfall (r) could be assumed as 200 mm based on the studied results of rainfall data analysis for Shenmu area from Lo $et\ al.$ (2012). The above formula indicates that only accumulated rainfall of the watershed is necessary to estimate the total of the new landslide area. Therefore, it is the key



to rainfall distribution characterization in typhoon events and data accuracy. It also affects the
regression analysis reliability. This study adopted Uchihugi formula to analyze correlation
between event-based landslides inventory and associated accumulated rainfall with its
corresponding new landslide ratio for predicting landslide magnitude in the future.
**4   Results**
This section utilized a dataset of complete and reliable landslide inventory maps of Shenmu
area from the 1996 Herb to 2009 Parma through the suggested image interpretation and
identification procedure which have edited and filtered out any possible error area for further
minimizing omissions. Also, it would help to effectively grasp the total landslide area and
new landslide area caused by an event in the study area and deeply explores the relative
relationship between landslide inventories and the environmental factors or rainfall events
with its corresponding accumulated rainfall. The main purpose of this analysis can be
concluded as:
a.  Temporal analyses: Using complete landslide inventory maps of the Shenmu area to
represent continuous landslide change with time subjected to each typical typhoon events
especially for most landslide-proneness of Chushui and Aiyuzih subwatersheds and also
recognize which event dominates maximum landslide area in the disaster history. In
addition, the range of the maximum and minimum landslide area would be constructed
based on the above analyzed results, which will help to quickly assess the effects of
landslide changes in the environment of an event in the future, whether or not fall within
acceptable limits to avoid landslide risk.
b.  Spatial analyses: Using spatial and statistical analysis with environmental factors or
rainfall events with its corresponding accumulated rainfall to find main landslide
contribution of each disaster events and next elaborate a combination of causative factors
for any given area within the study area. According to a combination of causative factors ,
one could develop a set of reason-based rules for preliminary delineation of potential
landslide area focusing on areas which have well-vegetated land cover and presently have
no evidence of landslide activity and situated at stable condition to lower damage loss of
disaster and implement rapid response and urgent decision-making during disaster.




As the above mentioned, this study completed the landslide analysis of Chushui and Aiyuzih
subwatersheds based on the Eq. (1) to Eq. (4) to represent the characteristic of spatial-
temporal distribution of landslide.
**4.1 Temporal Landslide Distribution**
Landslide distribution of Chushui and Aiyuzih subwatershed (see Fig. 8~9) in temporal scale
are separated into three periods to discuss typhoon/heavy rainfall-induced disaster magnitude
and its change after earthquake event.
**(1) Chushui subwatershed**
a. pre-1999 Chi-Chi earthquake events:
Before 1999 Chi-Chi earthquake, Chushui subwatershed had been subjected to extreme
rainfall events including 1996 typhoon Herb and 1999/5/28 heavy rainfall which caused
numerous landslide areas in the upstream watershed (Chen *et al.*, 2014) as shown in Fig. 10.
For landslide distribution in Chushui subwatershed during pre-1999 Chi-Chi earthquake
period, it is found that total landslide area ranged from 3.148 to 16.164 ha and its
corresponding *LR* is 0.365% to 1.876% before 1999 Chi-Chi earthquake. And, the new
landslide area caused by each event ranged frome from 9.372 to 13.973 ha and its
corresponding *NLR* is 1.088% to 1.622%. In this period, the study found that most landslide
area and its distribution was strongly affected by typhoon Herb (see Fig. 10) which brought
the highest accumulated rainfall up to 879mm with prolonged duration and also led to
frequent debris flow occurrence.
b. from1999 Chi-Chi earthquake to pre-typhoon Morakot events
In this period, Chushui subwatershed had been subjected to extreme rainfall events including
2001 typhoon Toraji, 2004 typhoon Mindulle, 2008 typhoon Kalmaegi, typhoon Fung-wong,
typhoon Sinlaku, typhoon Jangmi which reactivated old landslide areas and activated new
landslide areas in the upstream watershed as shown in Fig.10. For Landslide distribution of
Chushui subwatershed in the period between 1999 Chi-Chi earthquake and pre-typhoon
Morakot events, it is observed that total landslide area ranged from 14.465 to 57.196 ha and
its corresponding *LR* is 1.679% to 6.639%. And, the new landslide area caused by each event
ranged from 3.918 to 35.987 ha and its corresponding *NLR* is 0.455% to 4.177%. In this



period, one finds that most landslide area and its distribution are caused by 2004 typhoon
Mindulle (see Fig. 10). Then, compared this period with the previous, the amount of total
landslide and new landslide area after 1999 Chi-Chi earthquake is at least increased by 3.53
times and 2.57 times approximately. This fact implies that the main cause of the expanding
landslide area was the disturbance of geomaterial by strong earthquakes with giant seismic
shaking force (Lin *et al*., 2008).
c. post-typhoon Morakot events
In this period, Chushui subwatershed had been subjected to extreme rainfall events including
2009 typhoon Morakot and typhoon Parma. For landslide distribution of Chushui
subwatershed in post-typhoon Morakot events period, it is observed that total landslide area
ranged from 69.381 to 72.529 ha and its corresponding *LR* is 8.053% to 8.418%. And, the
new landslide area caused by each event ranged from 5.618 to 30.983 ha and its
corresponding *NLR* is 0.652% to 3.596%. Typhoon Morakot brought unpredictable
accumulated rainfall (2,099.5mm) with prolonged duration and high hourly intensity, this
event stuck in Chushui subwatershed and triggered many landslides including some large-
scale landslide over 10 ha and also caused debris flow occurrence and natural dams which
resulted in biggest manigtude of total landslide area within its disaster history.
**(2) Aiyuzih subwatershed**
a. pre-1999 Chi-Chi earthquake events
Before 1999 Chi-Chi earthquake, Aiyuzih subwatershed had been subjected to extreme
rainfall events as the same as Chushui subwatershed as shown in Fig.xx. In this period, total
landslide area ranged from 10.091 to 17.575 ha and its corresponding *LR* is 2.519% to
4.387% before 1999 Chi-Chi earthquake. And, the new landslide area caused by each event
ranged frome from 5.167 to 10.175 ha and its corresponding *NLR* is 1.290% to 2.540%. It is
obviously inferred that that most landslide area and its distribution was deeply affected by
1998/5/28 heavy rainfall (see Fig. 11) which does not coincide with the typical event of
Chushui subwatershed. Also, the total landslide area of Aiyuzih subwatershed in this period is
slightly over Chushui subwatershed about 2.275 ha even if Aiyuzih subwatershed area is
smaller than Chushui subwatershed.
b. from1999 Chi-Chi earthquake to pre-typhoon Morakot events



In this period, the amount of total landslide area Aiyuzih subwatershed increased significantly
after the 1999 Chi-Chi Earthquake about 32 ha. And, the landslide distribution of Aiyuzih
subwatershed in this period ranged from 32.276 to 57.693 ha and its corresponding *LR* is
8.056% to 14.4%. And, the new landslide area caused by each event ranged from 3.027 to
24.528 ha and its corresponding *NLR* is 0.756% to 6.122%. In this period, most landslide area
occurred at the river source which of this mass of sediment creates a direct supply of sediment
material to the river and soon caused debris flows induced by 2004 typhoon Mindulle and
2008 typhoon Kalmaegi subsequently. Comparing this period with the previous period, the
total landslide area and new landslide area after 1999 Chi-Chi earthquake increased by 3.28
times and 2.4 times approximately. Again, the 1999 Chi-Chi earthquake also plays a key
component in landslide activity affecting all of rainfall event-triggered landslide distribution
including its number and magnitude in Aiyuzih subwatershed (see Fig.11).
c. post-typhoon Morakot events
In this period, several large-scale landslide occurred after 2009 typhoon Morakot, most were
concentrated on the right river flank of the upstream watershed and oriented-north. The 2009
typhoon Morakot caused the new landslide area to increase to 86.590ha (*NLR*=21.613%) and
the total landslide area totals 133.036ha (*LR*=34.097%) which is more than a third of Aiyuzih
subwatershed area and it is indeed a greatest increase in total landslide area within its disaster
history as well as Chushui subwatershed.
**4.2    Spatial Landslide Distribution**
Landslides are generally a natural accompaniment of the geological cycles of uplift,
weathering and erosion. These long-term preparatory factors for landslides and the more local,
much shorter-term effects would trigger a particular failure (Hutchinson, 1995). To evaluate
all of the rainfall-induced landslide distribution in spatial scale for Chushui and Aiyuzih
subwatersheds, this study superimposed insights on geomorphology, geology, hydrology, and
human activity with regard to environmental and triggered factors. Accordingly, each unique
condition units (UCU) of landslide contribution (*LC*) was calculated to represent which
factors occupied most landslide distribution spatial scale and most possibly encourage future
landslide occurrence.



**(1) Geomorphology:elevation, slope and aspect**
A geomorphological map is used to depict elevation, slope gradient and aspect generated by
Digital Elevation Models (DEM) through a raster (grid) dataset of elevations. The elevation of
a mountain usually refers to its summit or divide which reflects climatic characteristics such
as temperature change and rainfall distribution. Landslides tend to occur on steeper slopes,
especially where the slope is covered by a thin colluvium. Aspect can influence moisture
retention and vegetation or drainage direction, which in turn can affect soil strength and
susceptibility to landslide (Chang *et al.*, 2007). So, this study use high-precision 5mx5m
DEM of the Shenmu watershed (see Fig.12) to derive geomorphology maps of elevation,
slope (in degrees) and aspect (see Fig.13~14) to assess landslide distribution in spatial scale
for each event. Figure 13 represents elevation of Chushui and Aiyuzih subwatersheds ranged
from 1000-3000m and 1000m-2500m. Both of the two elevation maps are divided into six
intervals by 500m spacing units for statistics of landslide distribution change affected by the
Chi-Chi earthquake. The results show landslide contribution (*LC*) of Chushui subwatersheds
subjected to 1996 typhoon Herb and 1999/5/28 heavy rainfall before Chi-Chi earthquake
concentrated in the interval of 2000-2500m elevation, which ranged from 42.51 to 54.16%
and averaged approximately 49.27%. After the Chi-Chi earthquake, the landslide contribution
(*LC*) affected by extreme rainfall events has dropped to 29.03%. Highest landslide
contribution (*LC*) belonged to the interval of 1500-2000m which ranged between 43.60 to
63.19% and averaged 52.38% with an increase by 1.21 times compared with the value before
the Chi-Chi earthquake. After 2009 typhoon Morakot, the highest landslide contribution still
belongs to the interval of 1500-2000m elevation about 53.08%. Consequently, it can be
deduced that typhoon-induced landslide potential in Chushui subwatersheds have gradually
moved towards the downstream watershed area with time obviously after 1999 Chi-Chi
earthquake. On the other hand, landslide contribution (*LC*) of Aiyuzih subwatersheds mainly
belonged to the interval of 1500-2000m which averaged close to 58% whether before and
after the Chi-Chi earthquake. (see Fig.15~16).
In terms of slope, according to Taiwan technical regulations for soil and water conservation
(SWCB, 2006), slopes are divided into seven classes (see Table 3). In consideration to the
environmental factor of slope, landslide contribution (*LC*) of Chushui and Aiyuzih
subwatersheds was mainly concentrated in class VI and VII sloppes before and after the Chi-

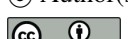


1 Chi earthquake. After typhoon Morakot, the average of two subwatersheds was between 65.58

2 to 83.31%. In addition, in consideration to the environmental factor of aspect, landslide

3 contribution (*LC*) of Chushui and Aiyuzih subwatersheds was mainly concentrated on west-

4 facing and southeast-facing slopes before and after the 1999 Chi-Chi Earthquake (see Fig.

5 15~16).

6 Namely, seismic effect did not affect the slope or aspect of geomorphological factor evidently,

7 but elevation under seismic effect has more dominated landslide distribution than the other

8 geormorphological factors in these watersheds.

9 **(2) Geology:Lithology**

10 Landslide occurrence influenced by geological factor is determined by the lithological

11 characteristics, formation and degree of weathering. If the geological formation is fractured,

12 high-permeability, exhibiting shallow soil layers and dense faults, this may influence severe

13 and frequent landslides in subsequent heavy rainfall events (Chen *et al*., 2014). Except the

14 few downstream alluvial places, most geological setting of Chushui subwatershed is

15 Nanchuang Formation. Therefore, most typhoon/rainfall-triggered landslides areas are located

16 within Nanchuang Formation. The average landslide contribution (*LC*) reaches 99% before

17 and after the 1999 Chi-Chi Earthquake (see Fig. 15~16). In contrast, the geological setting of

18 Aiyuzih subwatershed is composed of Nanchuang Formation in upstream area and Hoshe

19 Formation in downstream area. The average landslide contribution (*LC*) of Nanchuang

20 Formation ranged between 71.66 to 82.19% and is greater over Hoshe Formation before and

21 after the 1999 Chi-Chi Earthquake (see Fig. 15~16).

22 Lin *et al*. (2008) has studied on typhoons and earthquakes on rainfall-induced landslides in

23 central Taiwan and also found that landslide distribution is intimately related with the uniaxial

24 compressive strength. They concluded that the average uniaxial compressive strength of

25 Nanchuang Formation, at 42 MPa, is lower than the average uniaxial compressive strength of

26 Hoshe Formation and the increase in the landslide ratio of Nanchuang Formation is higher

27 than the increase of Hoshe Formation which also agree well with the above observations.

28 **(3) Hydrology:river erosion and rainfall**

29 The influence of river erosion on the landslide phenomena activation is expressed in the

30 transportation of the sediment material from lower retaining parts of the slopes and





disturbance to the slope equilibrium. The lateral erosion (toe-cutting) is observed mainly in
rivers with constant water current. The activity of the rivers bearing their own sediments,
when changes in the water level occur, easily leads to rapid alternations in the position of
riverbeds or the offset of river course (Margottini *et al*., 2013). Consequemtly, it would
positively encourage the activation of landslide process especially when suffering exreme
rainfall events. For evaluating landslide correlated with river erosion, a buffer zone was set to
external expansion 25m on both sides of the river system for statistical analysis of landslide
contribution (*LC*) of Chushui and Aiyuzih subwatersheds (SWCB, 2010[a]). Figure 15~16
shows most typhoon/rainfall-triggered landslides areas of the two subwatersheds are located
within a buffer zone of river system before and after the 1999 Chi-Chi Earthquake. After the
1999 Chi-Chi Earthquake, landslide contribution of both watersheds affected by river erosion
is significantly increasing year upon year and has reached the peak after the 2009 typhoon
Morakot. Landslides is discernibly identified along river courses with the two watersheds
especially in the past ten years.
On the other hand, the modified Uchihugi formula based on Eq. (4) can be employed to depict
the relationship between the new landslide ratio of each event versus its corresponding
accumulated rainfall. Figure 17 represents accumulated rainfall increases which would indeed
enlarge the new landslide ratio of the two subwatersheds. It is obvious that the corresponding
accumulated rainfall is a major triggering factor to induce landslide occurrence during
typhoon season. It could be inferred that the new landslide ratio of Aiyuzih subwatershed is
averagely greater than Chushui subwatershed over 1.64 times subjected to the same
accumulated rainfall of pseudo event.
**(4) Human activity:land use and road construction**
Human activities with regard to land use such as the planting of crops, clearance of vegetation
or geotechnical engineering projects such as road construction have important effects on slope
instability (Cotecchia, 1978; Greenway 1987; Hutchinson, 1995). It is indicated that land use
belonging to bare land such as streams, canals and shoal of Chushui and Aiyuzih
subwatersheds have landslide contribution (*LC*) greater than 56% at least before and after the
1999 Chi-Chi Earthquake  (see Fig. 15~16). Also, these observations are fitted with the results
from the influence of river erosion on the landslide which infers majority of typhoon/rainfall-
triggerred landsides are activated or reactivate close to the river courses.



Similarly to evaluate landslide correlation with road construction, a buffer zone was set to
external expansion of 25m on both sides of the road network for statistical analysis of
landslide distribution of Chushui and Aiyuzih subwatersheds (SWCB, 2010[a]). Since no road
passes through Aiyuzih subwatershed, the landslide contribution is zero. Moreover, Highway
18 has passed through the south of Chushui subwatershed and the average landslide
contribution only reaches from 3.76~4.85% with a buffer zone of road network. It is evident
that there is less relevant correlation with human activity. (see Fig. 15~16)
**5.    Discussion**
**5.1    Earthquake Amplification Effect**
Sediment-related disasters, including debris flow and landslides, have frequently occurred in
Taiwan during the past two decades, especically following the 1999 Chi-Chi earthquake ($M_L$ =
7.3). The most well-documented recent debris-flow events were those caused by typhoons,
including  2001 typhoon Toraji, 2004 typhoon Mindulle and 2009 typhoon Morakot (Cheng et
al., 2005; Chang *et al*., 2007; Tsai et al., 2010; Wu et al., 2011; Lo et al., 2012; Chen, et al.,
2014). The Shenmu watershed was approximately 36 km from the epicentre of the 1999 Chi-
Chi earthquake, during which the peak ground acceleration (PGA) reached betwwen 250 to
400 gal (Chung, 1999). As the previously mentioned, spatial and statistical results based on
the complete landside inventory maps of both Chushui and Aiyuzih subwatersheds exhibited
that in temporal-scale distribution, the three major sediment-related disasters highlights a
significant increase of landslide ratio. Until October, 2009 after typhoon Parma, the statistics
shows the total landslide areas of Aiyuzih subwatershed was about 2 times more than Chushui
subwatersheds. That proves the strong seismic effect of 1999 Chi-Chi earthquake has
destructive impact on local geological condition due to the lithological strength and structure,
which can lead to the weakening of the cohesion and strength of rock and soil mass near
hillcrests, which in turn can lead to more or large-scale typhoon/heavy rainfall-triggered
future landslides occurrence after an earthquake (Havenith *et al*., 2006, Chang *et al*, 2007).
From a geographical point of view, both of Chushui and Aiyuzih subwatersheds located at
Alishan mountain belong to one of the top five highest mountains in Taiwan. Also, Aiyuzih
subwatersheds is closer in the proximity of the Chi-Chi earthquake epicenter with smaller area





and steeper terrain than Chushui subwatersheds. Geli *et al*. (1988) has found that earthquake-
triggered landslides have been related to the topographic characteristics as well as
amplification effect of topography, which means seismic motion is amplified at mountain tops.
In addition, the proximity to the earthquake epicentre and earthquake fault has been suggested
by Keefer (2000) as proportional to the surface area disturbed by landsliding and the increase
becomes more obvious with proximity to the epicenter. The landslide data of Chi-Chi
earthquake was highlighted by Dadson *et al*. (2004) whose findings stated that the decrease in
area affected by landslide away from the Chelungpu fault was rapid at distances in excess of
20 km from the fault (see Fig.18). Accordingly, it can be logically deduced that Aiyuzih
subwatersheds after 1999 Chi-Chi earthquake, the landslide ratio increased more obviously
than Chushui subwatersheds with proximity to the epicentre especially subjected to 2009
typhoon Morakot.
**5.2   Combination of Causative Factors**
Through the obtained results from the previous spatial and statistical analysis with
environmental factors related to landslide inventory, a combination of causative factors for
any given area within the study area can be elaborated. Figure 19 represents the average
landslide contribution (*LC*) of various environmental/causative factors after 1999 Chi-Chi
earthquake for Chushui and Aiyuzih subwatersheds. These causative factors are also
subdivided into natural and anthropogenic factors. Among them, anthropogenic factors (as
well as human activity including land use and road construction) cause minor or irrelevant
landslide contribution in the two subwatersheds, but natural factors (including elevation, slope
gradient, lithology and river erosion) dominate landslide contribution within the study area.
This means these two areas have infrequent human interference and landslide process is only
controlled by typhoon/heavy rainfall events during flood season. Afterward, the combination
of major relevant causative factors would be arranged to set a series of logic reason-based rule
due to the studied results as shown in Fig. 20~22. The attributes of highest-potential landslide
location are concluded to be situated on 1500m ~2000m of elevation which of slope gradient
over 55% and oriented west/southearst, lies in Nanchuang Formation and adjacent to a 25m
buffer zone of river course. Furthermore, using logic reason-based rules can quickly detect or



assess the future landslide locations and then can serve as the basis for effectively managing
watersheds, planning of disaster prevention works and reducing risk of landslide in advance.
**5.3   Landslide Potential Map**
The contribution and functional relationship between various factors (environmental and
triggering) affecting slope instability, the spatial-temporal distribution of landslides and the
prediction of landslide occurence has been highlighted in many studies throughout the past
century and has gained increased scrutiny from researchers in recent years (Guzzetti et al.,
1999; Ayalew and Yamagishi, 2005). This study utilized the logic reason-based rules based
on a combination of geomorphology, geology, hydrology and other environmental factors to
assess and delineate the future landslide potential area of the two watersheds. The generally
accepted correlation between past and future factors leading to slope instability or leading to
landslide occurence has been highlighted by previous researchers (Clerici et al., 2006). The
results revealed that areas which have well-vegetated land cover presently have no evidence
of landslide activity (ie. there are no signs of collapse or active areas). Based on the
assumption that impact of triggerring factors are excluded, landslide potential of the two
watersheds have been assessed, categorized, and mapped into three classes. These classes
constitute a clear and definite representation of the relative levels of future landslide
occurrence threat. The future landslide potential map of Chushui and Aiyuzih subwatersheds
are composed of three classes as follows: low, moderate and high. In practical applications,
the low potential landslide areas fit any one of the developed three logic reason-based rules
based on a combination of environmental factors as show in Fig. 20. Secondly, the moderate
potential landslide areas fit any two of the developed three logic reason-based rules based on a
combination of environmental factors as show in Fig. 21. Finally, the high potential landslide
areas fit all of the developed three logic reason-based rules based on a combination of
environmental factors as show in Fig. 22.
According to Fig. 23, the landslide potential map of Chushui and Aiyuzih subwatersheds are
regularly delineated and then generated qualitatively. Table 4 lists statistics of three classes
potential of landslide area within the two subwatersheds. The landslide potential area of
Chushui subwatersheds still exists about 721.11 ha which involved 112.3 ha of moderate and
high potential landslide area occupied in 13.24% of the whole watershed. And, the landslide





potential area of Aiyuzih subwatersheds still exists about 231.11 ha which involved 31.7 ha of
moderate and high potential landslide area occupied in 2.72% of the whole watershed.
Comparing both subwatersheds, high potential landslide area of Chushui subwatersheds is 3.3
ha more than Aiyuzih, and the largest part of areas are 1.53 ha located at right flank of the
middle river portion. In contrast, the high potential landslide area of the Aiyuzih subwatershed
have occurred mostly in past so that there only rests 0.1 ha of the high potential landslide area
located at the left flank of upstream subwatershed adjacent to river source.
From a risk assessment point of view, investigations of elements at risk in the two
subwatersheds conclude the native population who live, work and travel through the area
suffer and has limited number of properties such as houses and buildings are the greatest
exposed to landslide hazard. Landslide risk to these elements at risk is generally seen as low
within the two subwatersheds. However, some residents of Shenmu Village at present still
live close to the confluence of two subwatershed where Highway 21 passes through the
Shenmu Bridge and the Elementary School constructed nearby. Therefore, the instantaneous
evolution of moderate and high potential landslide area should be paid more attention to and
periodically monitored depending on remote sensing images (satellite/radar images and aerial
photos) or using the long-term observation stations equiped with rainfall gauges, geophones
and others especially during the typhoon season and then conduct rapid decision-making
process to set sophisticated engineering measures or implement rapid evacuation responses
for effectively reducing damages, loss of life and risk from the debris-flows disaster caused by
the potential landslide area under extreme rainfall condition.

## 6.   Conclusion

Landslide and other sediment-related disasters are a natural phenomena related to the cycle of
land degradation which affects many populations world-wide. In Taiwan, landslides are
frequent occurences especially during extreme events such as earthquakes and typhoons.
Under the effect of global climate change, the probability of extreme weather occurrence has
increased. These events act as triggering factors for landslides on unstable slopes. Therefore,
landslide frequency in Taiwan has seen an increase over time. The cycle of land degradation
and the high uncertainty with recurrent characteristics of landslides has led to the formation of
vast amounts of unstable sediment material deposited on hillslopes which could easily lead to



other secondary disasters (eg. landslide dams and debris flows). Furthermore, landslide
processes are a complicated mechanism which is further complicated by the various triggering
factors combined with causative factors (eg. environmental factors) depending on the local
characteristics of specific region or given watershed scale. Through spatial analysis of multi-
event landslide inventory maps, it can be deduced that the estimated/predicted potential
landslide area inherits a combinations of causative factors of the historical landslide
experienced in the watershed/slope. Then, the landside inventory maps from 1996 typhoon
Herb to 2009 typhoon Parma is established systematically and exhaustively by semi-
automated image interpretation and artificial image identification procedures for ensuring the
completeness and accuracy of the dataset composing 11 historical disaster event involving 17
satellite images set. This study used the proposed methods based on the developed event-
based landside inventory maps to analyze the overall landslide evolution, magnitude of
landslide, landslide location and landslide potiental affecting by these extreme events in time
and space domains. Also, the strong seismic effect of 1999 Chi-Chi earthquake and the
quantified magnification of this event on amount of landslide for subsequent typhoon events
were included in this study. Finally, the studied results and concrete observation can be
concluded as follows:
1. The study suggests a framework for semi-automated image interpretation and artificial
image identification procedures which dramatically improves the quality of data content
and data structure so that it can definitely promote data integrity and accuracy for
improving value and reliability of event-based landslide inventories. For ensuring data
integrity, it is necessary to utilize high-resolution satellite image set close to the date of
event occurrence and also cross check the collected historical aerial photographs and local
landslide reports in order to reduce the influence of artefacts which result in data
imperfection or deficiency of landslide inventory such as cloud cover after the disaster
event or shadowing which leads to omission of the interpreted landslide polygon subject.
For controlling data accuracy, image registration should first be conducted for all of the
satellite image in order to reduce spatial location shifting of image causing false positives.
In addition, the land use/land cover map and aerial photographs are used to eliminate
likely-landslide polygon due to human activities for extracting the real landslide area
caused by natural hazard. The above framework can be applied to similar research to





study the completeness of landslide inventory with enviromental factors for any given
area.
2. The results of temporal scale analysis of the landslide inventories are used to classify the
event which caused the highest increase in landslide area among the three designated
periods. Before the 1999 Chi-Chi earthquake, the extreme events with the greatest impact
on Chushui and Aiyuzih subwatersheds were regarded as 1996 typhoon Herb and
1999/5/28 heavy rainfall respectively. From 1999 Chi-Chi earthquake to pre-typhoon
Morakot, the extreme events with the greatest impact was 2004 typhoon Mindulle. In
view of the post-typhoon Morakot events, it is obvious that 2009 typhoon Morakot
resulted in the greatest impact on the two watersheds with a large increase in landslide
area. After comparisons were made of the total landslide and new landslide area before
and after 1999 Chi-Chi earthquake in the two subwatersheds, it was found that the
earthquake amplification effect of the quantified magnification was estimated to be at
least doubled that agrees well with the previous studies. Furthermore, from the
relationship between the new landslide ratio of each event versus its corresponding
accumulated rainfall, the modified Uchihugi formulas of two subwatersheds could be used
to quickly estimate new landslide area caused by of a given event during typhoon season
depending on the parameter of accumulated rainfall. Estimation of the new landslide area
caused by an event can provide useful information as a reference basis for preliminary
evaluation of the exposure of elements at risk downstream for early warning , rapid
response, urgent evacuation and longterm mitigation solutions.
1. According to the long-term average trend of landslide contribution after the Chi-Chi
earthquake events, it can represent which environmental factors are major relevent
causative factors and be further utilized to establish as a series of logic reason-based rules
to delineate potential landslide areas. Accordingly, the generated landslide potential map
based on the developed logic reason-based rules may be as simple as a semi-quantitative
map that can be used to display the locations of old and new landslides to indicate
potential instability. For moderate-high potential landslide area, it is recommended to
construct preventive engineering measures by its necessity and requirement of
remediation or utilize education for effectively reducing loss of life and damage to public
property and  making landslide risks more tolerable and generally acceptable limit.



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



Table 1. Historical rainfall events and its corresponding rainfall data

| Date | Rainfall event | $R_A$ (mm) | | Maximum rainfall intensity (mm/h) | | Duration (hr) | | Debris flow occurrence |
|---|---|---|---|---|---|---|---|---|
| | | C | A | C | A | C | A | |
| 1996/07/29 – 08/01 | Herb | 879 | 1987 | 70 | 112.5 | 96 | 173 | C |
| 1999/5/27-5/28 | 528 Heavy rainfall | 284 | 313.5 | 21.5 | 23.5 | 48 | 24.5 | C |
| 2001/07/28 – 31 | Toraji | 587 | 757 | 75.5 | 121 | 96 | 82.5 | C |
| 2004/06/28 – 07/03 | Mindulle | 816.5 | 1181.5 | 45 | 84.5 | 144 | 10 | A,C |
| 2008/07/16 – 07/18 | Kalmaegi | 529.5 | 619 | 61.5 | 80 | 72 | 55 | A,C |
| 2008/07/26 – 07/29 | Fung-wong | 500 | 641.9 | 35 | 50 | 96 | 48 | - |
| 2008/09/11 – 09/16 | Sinlaku | 922 | 1470.5 | 30.5 | 53 | 144 | 83 | - |
| 2008/09/26 – 09/29 | Jangmi | 648 | 885.5 | 39.5 | 64.5 | 96 | 75.5 | - |
| 2009/08/05 – 08/10 | Morakot | 2099.5 | 3060.5 | 72 | 123 | 144 | 28.5 | A,C |
| 2009/10/03 – 10/06 | Parma | 18 | 154 | 3.5 | 31.5 | 96 | 19 | - |

Note 1: The A and C are referred to be Aiyuzi and Chushui subwatershed, respectively.
Note 2: $R_A$ means accumulated rainfall of a given event.
Table 2. Correspondence between Historical Hazard Events and Satellite Images

| NO. | Rainfall event (Date) | Satellite image sets | | | |
|---|---|---|---|---|---|
| | | Stage | Date | Sensor | Resolution |
| 1 | Herb (1996/07/29–08/01) | pre-typhoon Herb | 1996/04/17 | SPOT-3 | 6.25 m |
| | | post-typhoon Herb | 1996/11/08 | SPOT-2 | 6.25 m |
| 2 | 528 Heavy rainfall (1999/5/27-5/28) | pre-528 heavy rainfall | 1999/03/26 | SPOT-4 | 6.25 m |
| | | post-528 heavy rainfall | 1999/07/24 | SPOT-2 | 6.25 m |
| 3 | Chi-Chi Earthquake (1999/09/21) | post-earthquake Chi-Chi | 1999/10/31 | SPOT-4 | 6.25 m |
| 4 | Toraji (2001/07/28–31) | pre-typhoon Toraji | 2001/01/20 | SPOT-4 | 10 m |
| | | post-typhoon Toraji | 2001/10/22 | SPOT-4 | 10 m |
| 5 | Mindulle (2004/06/28–07/03) | pre-typhoon Mindulle | 2004/02/10 | SPOT-5 | 2.5 m |
| | | post-typhoon Mindulle | 2004/07/10 | SPOT-5 | 2.5 m |
| 6 | Kalmaegi (2008/07/16–07/18) | pre-typhoon Kalmaegi | 2008/07/05 | SPOT-5 | 2.5 m |
| | | post-typhoon Kalmaegi | 2008/07/21 | SPOT-5 | 2.5 m |
| 7 | Fung-wong (2008/07/26–07/29) | post-typhoon Fung-wong | 2008/08/26 | SPOT-5 | 2.5 m |
| 8 | Sinlaku (2008/09/11–09/16) | post-typhoon Sinlaku | 2008/09/21 | SPOT-5 | 2.5 m |
| 9 | Jangmi (2008/09/26–09/29) | post-typhoon Jangmi | 2008/11/12 | SPOT-5 | 2.5 m |
| 10 | Morakot (2009/08/05–08/10) | pre-typhoon Morakot | 2009/05/08 | SPOT-5 | 2.5 m |
| | | post-typhoon Morakot | 2009/09/04 | SPOT-5 | 2.5 m |
| 11 | Parma (2009/10/03–10/06) | post-typhoon Parma | 2009/10/21 | SPOT-5 | 2.5 m |





Table 3. Landslide Correlation Hazard-Inducing Factor Classification in Chushui and Aiyuzih
subwatersheds

| environmental category | Factor | Classification | |
|---|---|---|---|
| Geomorphology | Elevation (m) | 0≤ elevation <500<br>500≤ elevation <1000<br>1000≤ elevation <1500 | 1500≤ elevation <2000<br>2000≤ elevation <2500<br>2500≤ elevation <3000 |
| | Slope Gradient(%) | Class I: 0≤ slope <5%<br>Class II: 5%≤ slope <15%<br>Class III: 15%≤ slope <30%<br>Class IV: 30%≤ slope <40% | Class V: 40%≤ slope <55%<br>Class VI: 55%≤ slope <100%<br>Class VII: ≥100% |
| | Slope Aspect(°) | North(N)<br>Northeast(NE)<br>East(E)<br>Southeast(SE)<br>South(S) | Southwest(SW)<br>West(W)<br>Northwest(NW)<br>Flat |
| Geology | Lithology | Hoshe Formation | Nanchuang Formation |
| Hydrology | River Erosion | Set 25*m from the line to be the affected extent to compile landslide area within and not within this limit. | |
| | Rainfall | Typhoon Herb<br>528 heavy rainfall<br>typhoon Toraji<br>typhoon Mindulle<br>typhoon Kalmaegi | typhoon Fung-wong<br>typhoon Sinlaku<br>typhoon Jangmi<br>typhoon Morakot<br>typhoon Parma |
| Human Activity | Land Use | Artificlal Forest Land<br>Bared Land<br>Cropland and Pasture<br>Herbaceous Rangeland<br>Nature Forest Land | Residential<br>School<br>Shrub and Brush Rangeland<br>Streams and Canals<br>Transportation |
| | road construction | Set 25*m from the line to be the affected extent to compile landslide area within and not within this limit. | |

Note: slope classification system referred to SWCB (2006)
Table 4. statitics of low, moderate and landslide potential aread within Chushui and Aiyuzih
subwatersheds

| Subwatershed area (ha) | | Low potential landslide | | Moderate potential landslide | | High potential landslide | | Total | |
|---|---|---|---|---|---|---|---|---|---|
| | | Area (ha) | Occupied rate (%) | Area (ha) | Occupied rate (%) | Area (ha) | Occupied rate (%) | Area (ha) | Occupied rate (%) |
| Chushui | 848.24 | 609.6 | 71.86% | 109.0 | 12.85% | 3.3 | 0.39% | 721.9 | 85.1% |
| Aiyuzi | 397.51 | 220.3 | 55.41% | 10.7 | 2.70% | 0.1 | 0.02 | 231.1 | 58.13% |

Note: Ocupied rate (%) = (potential landslide area) / (subwatershed area)







2    Figure 1. Geographic Location and Geology map of Shenmu area


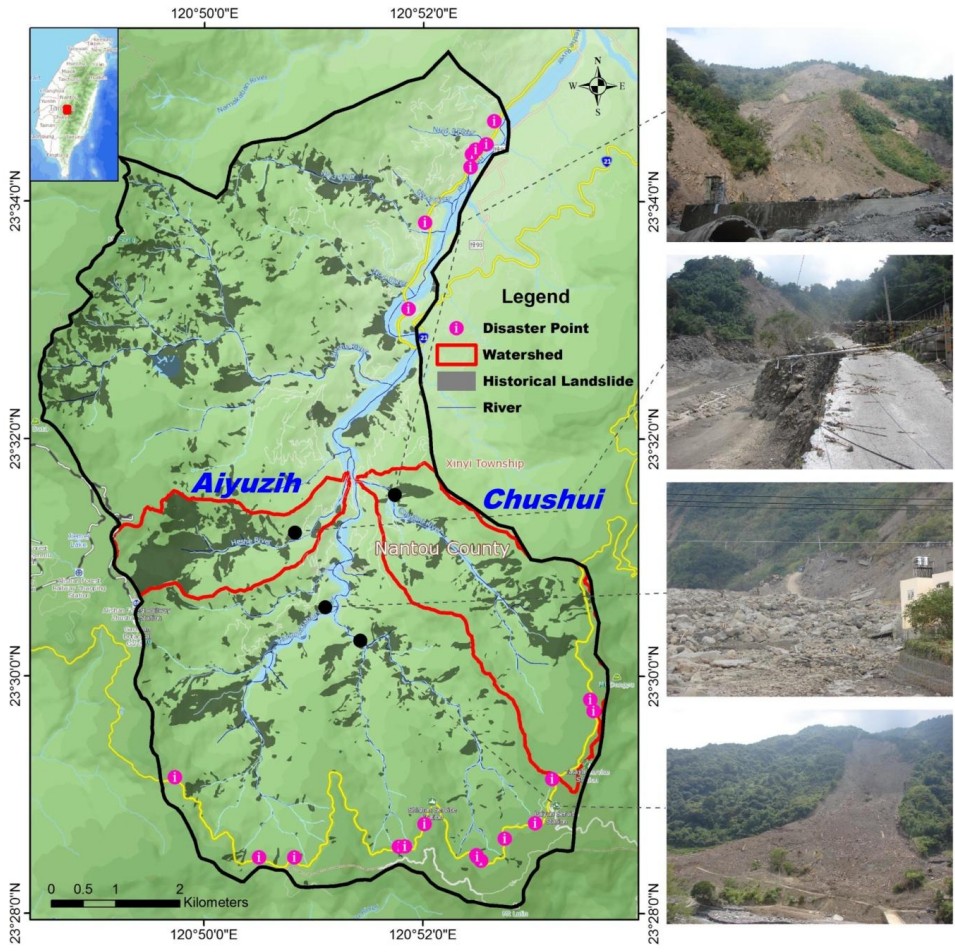

2    Figure 2 Landslide distribution and disaster points in Shenmu watershed after typhoon

3    Morakot



Figure 3. Path of Historical Typhoons which affected Taiwan.

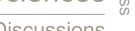
pre-typhoon
Herb(1996/04/17)

post-typhoon
Herb(1996/11/08)

pre-528 heavy
rainfall(1999/03/26)

post-528 heavy
rainfall(1999/07/24)

post-earthquake Chi-
Chi(1999/10/31)

pre-typhoon
Toraji(2001/01/20)

post-typhoon
Toraji(2001/10/22)

pre-typhoon
Mindulle(2004/02/10)

post-typhoon
Mindulle(2004/07/10)

pre-typhoon
Kalmaegi(2008/07/05)

post-typhoon
Kalmaegi(2008/07/21)

post-typhoon Fung-
wong(2008/08/26)

post-typhoon
Sinlaku(2008/09/21)

post-typhoon
Jangmi(2008/11/12)

pre-typhoon
Morakot(2009/05/08)

post-typhoon
Morakot(2009/09/04)

post-typhoon
Parma(2009/10/21)

1          Figure 4. Schematic layout of the adopted satellite images set





3    Figure 5. Procedure of the proposed image interpretation and identification





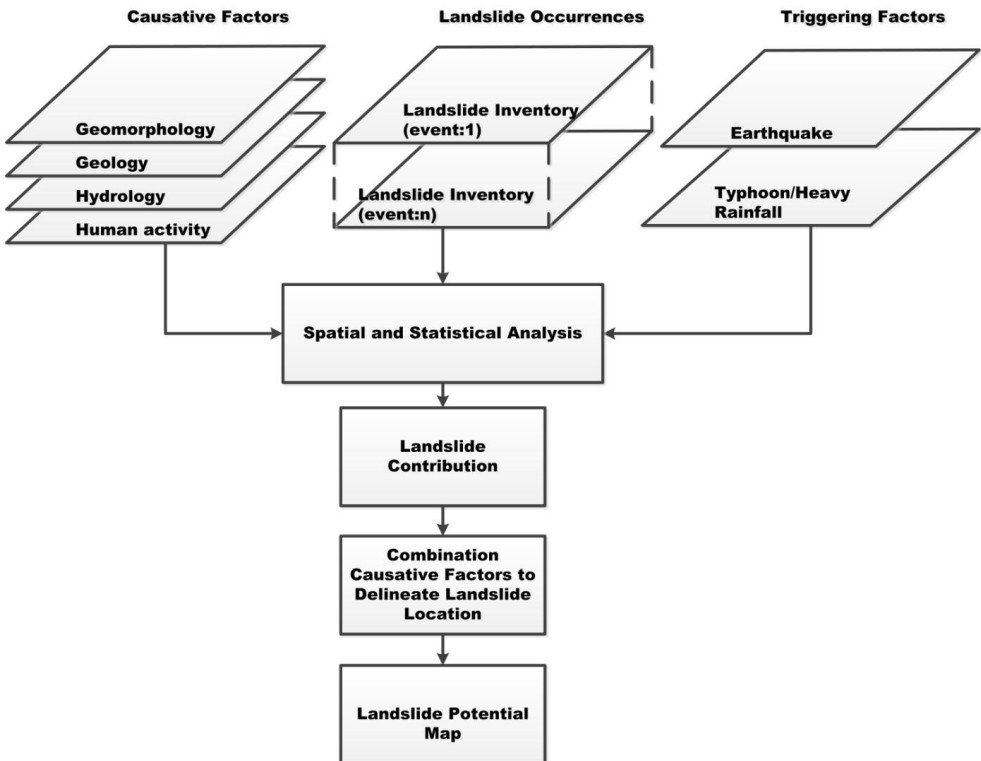

2    Figure 6. Procedure of spatial and statistical analysis environmental and triggered factors with

3        landslide inventory maps

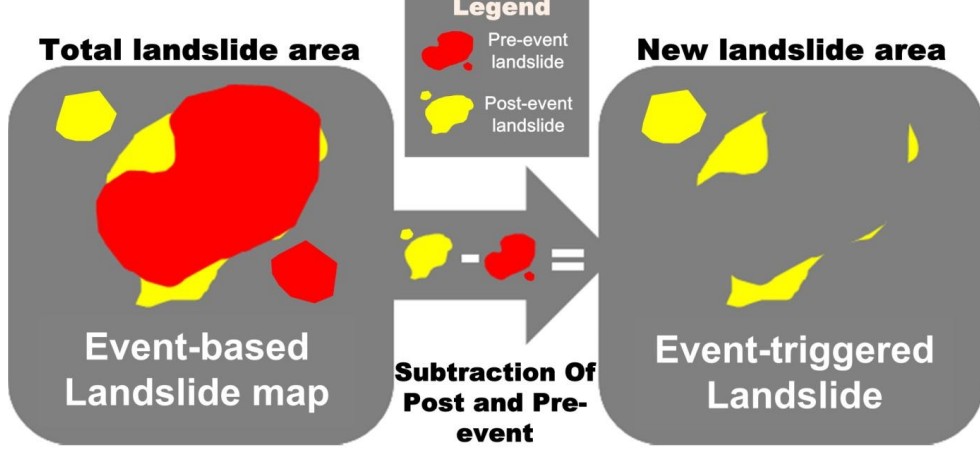

6    Figure 7.  Sketch of total and new landslides area definition





Figure 8. Landslide distribution of Chushui subwatershed for each event





Figure 9. Landslide distribution of Aiyuzih subwatershed for each event



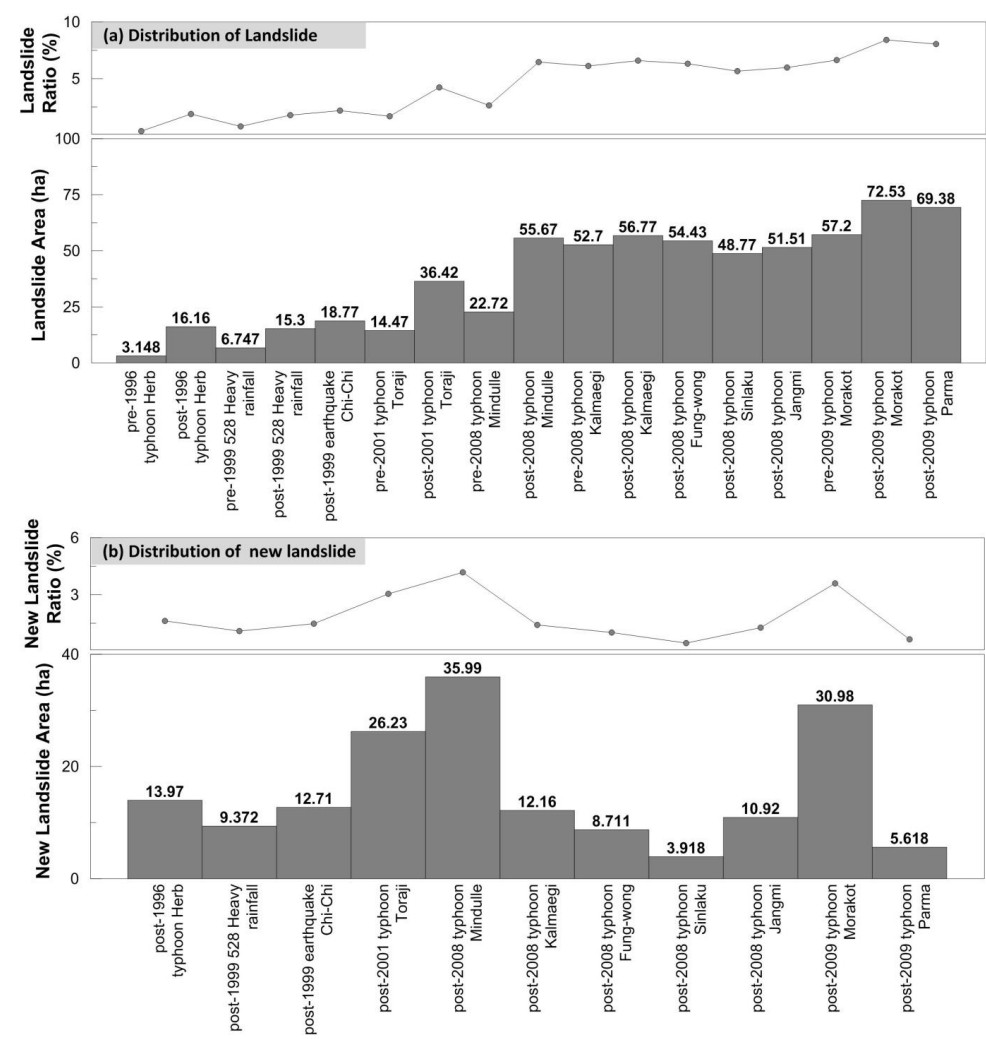

2    Figure 10. Landslide Distribution of Chushui subwatershed for each event in Temporal Scale




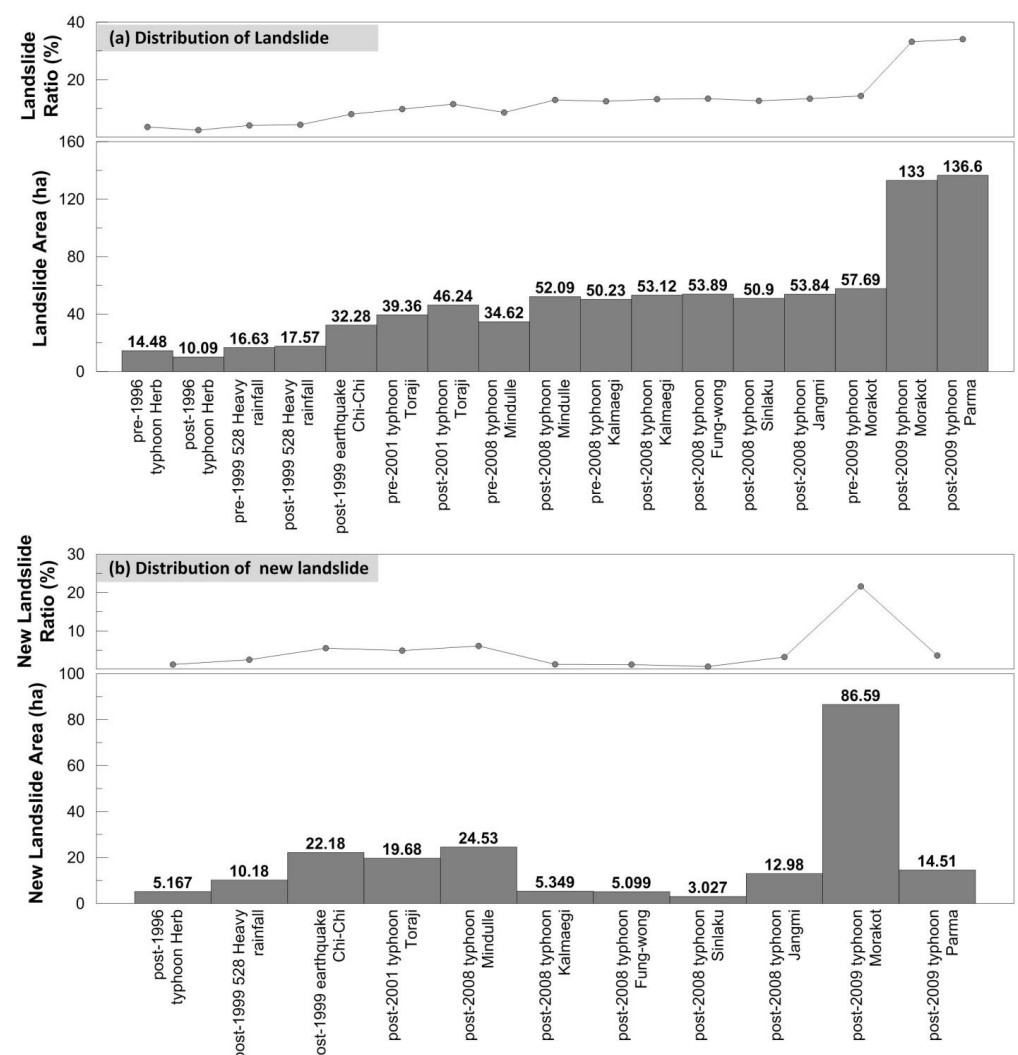

2  Figure 11. Landslide Distribution of Aiyuzih subwatershed for each event in Temporal Scale



2      . Figure 12. 5mx5m DEM and terrain relief of the Shenmu watershed





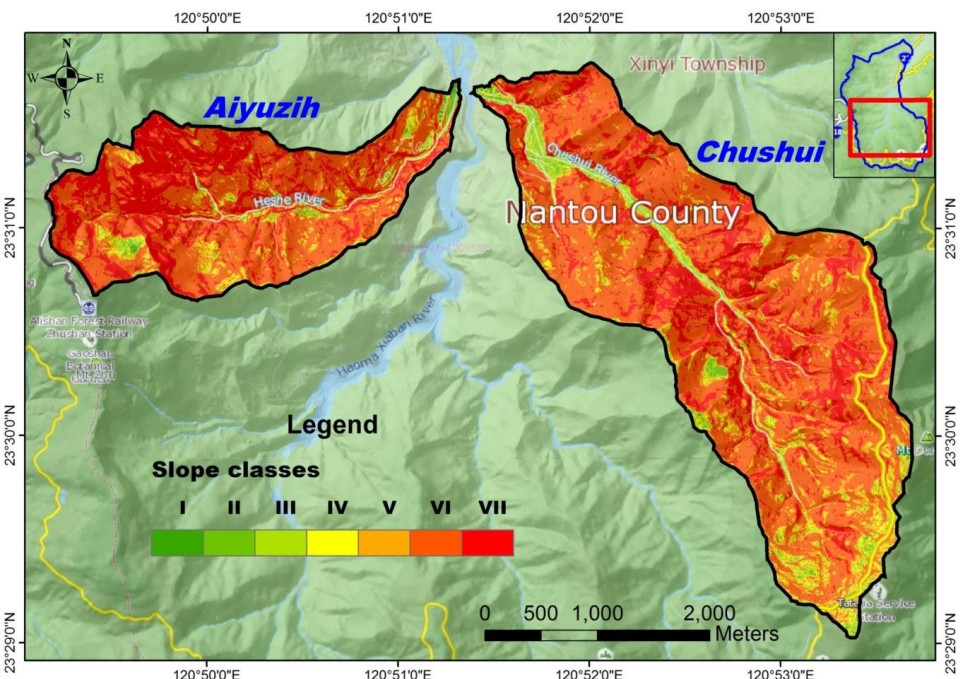

2    Figure 13. Slope map of Chushui and Aiyuzih subwatershed

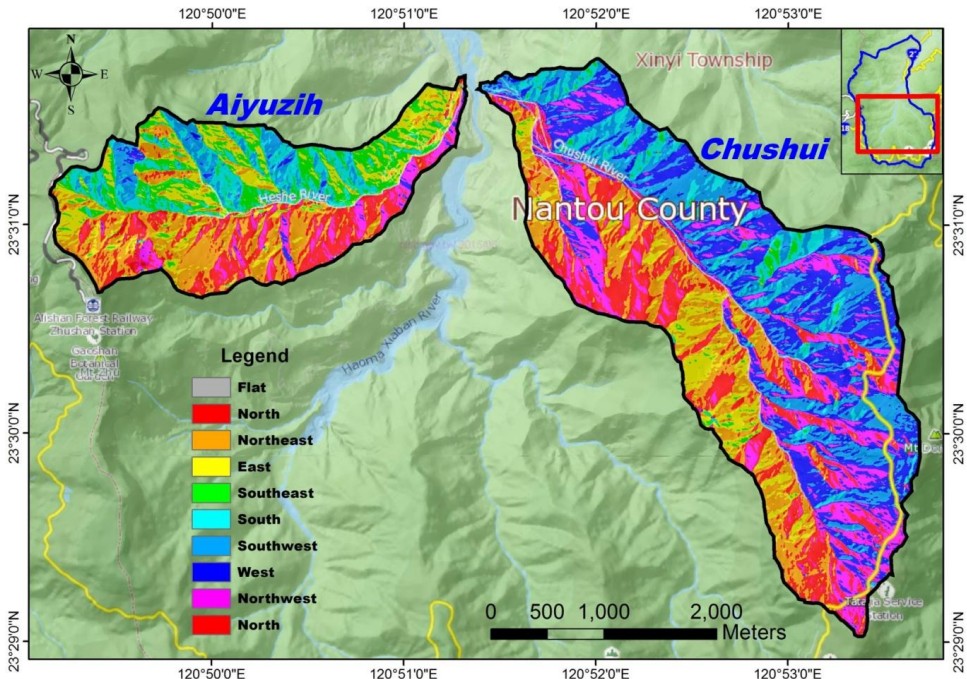

4    Figure 14. Slope aspect map of Chushui and Aiyuzih subwatershed




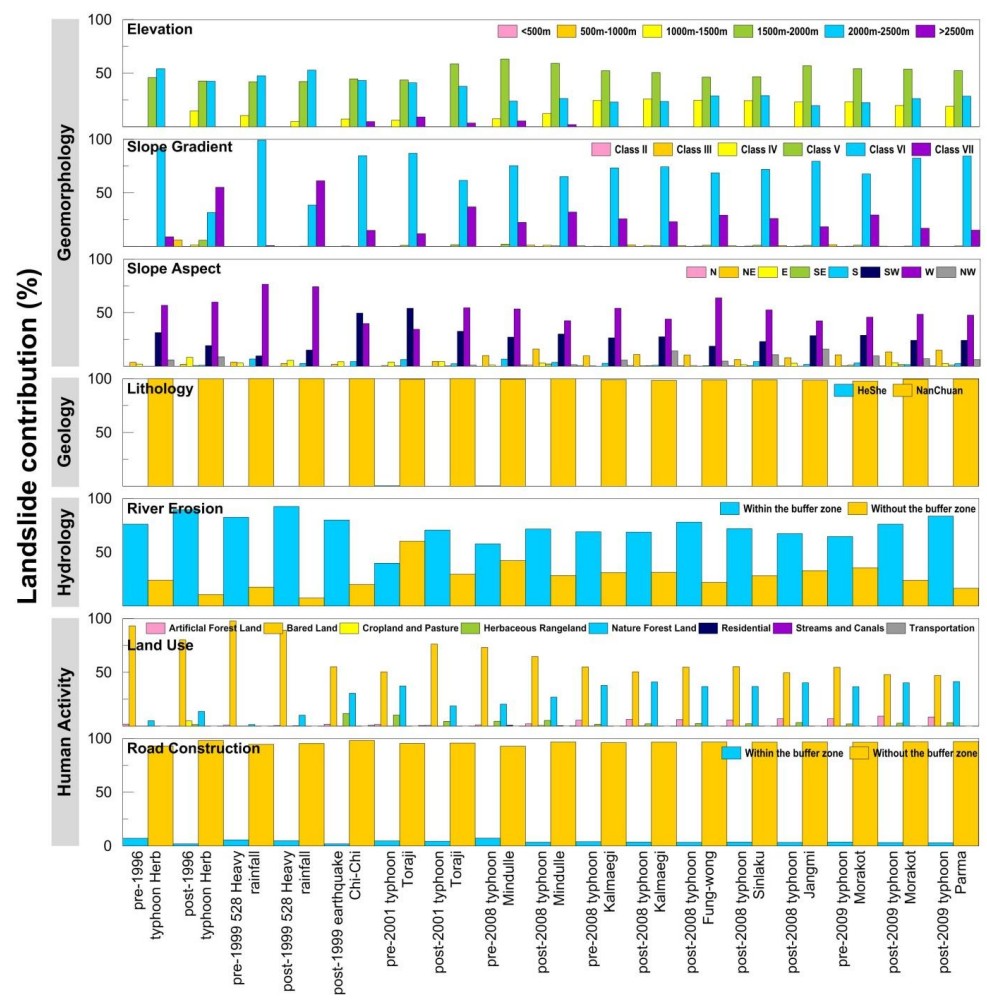

2   Figure 15. Landslide contribution (LC) of Chushui subwatersheds affected by environmental

3   factors





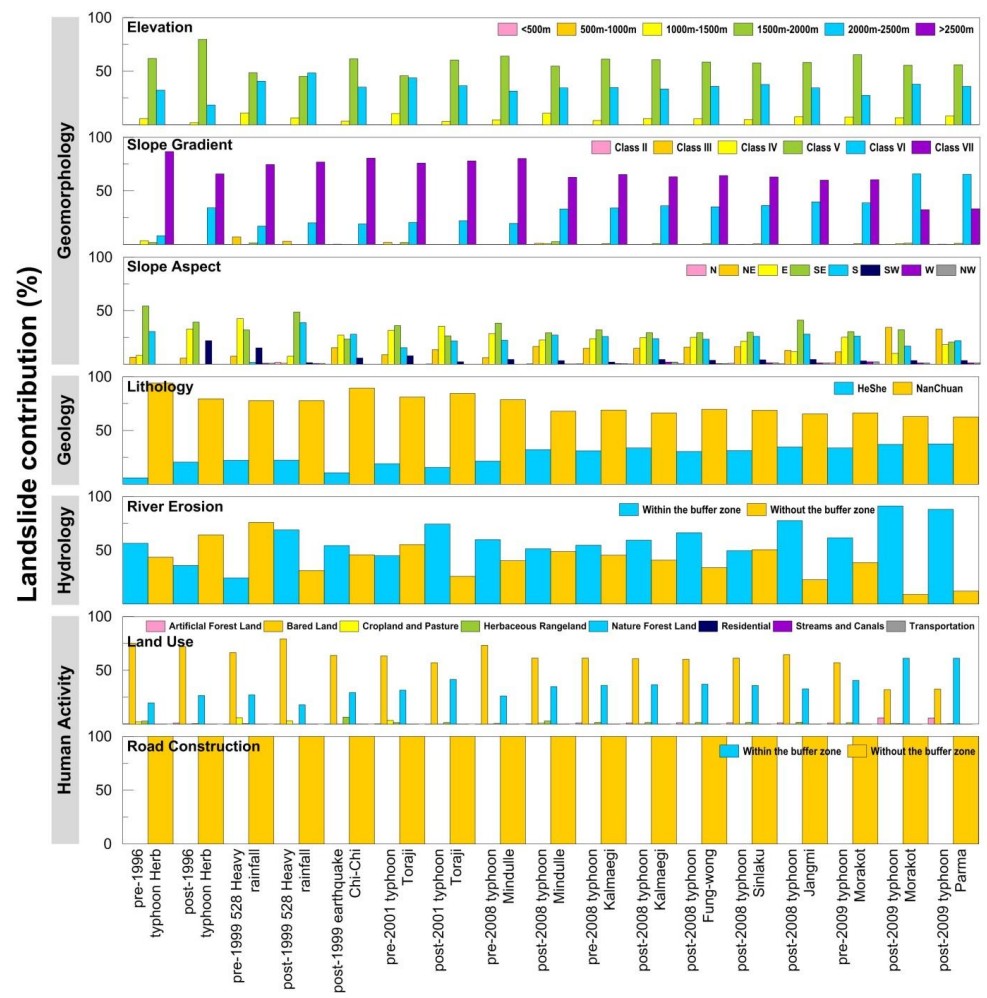

2    Figure 16. Landslide contribution (LC) of Aiyuzih subwatersheds affected by environmental

3    factors

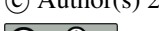


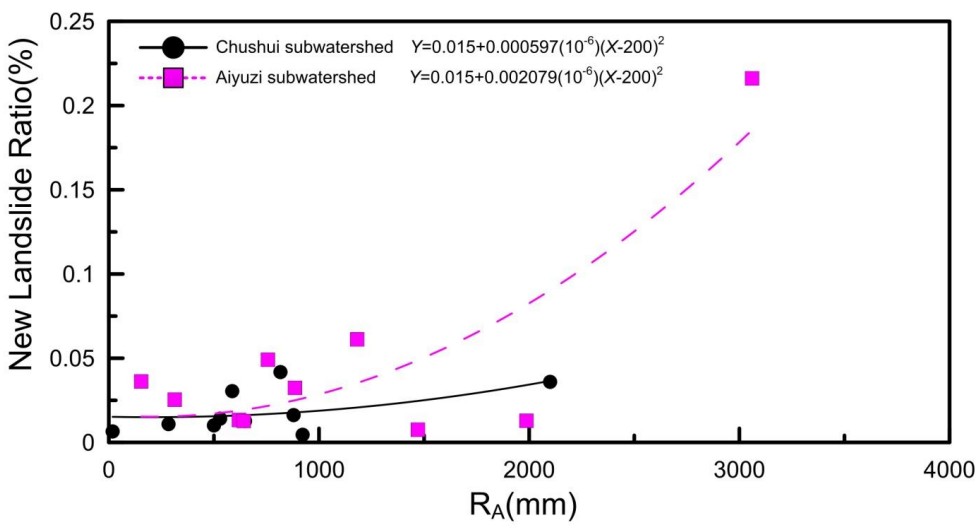

Figure17. Graph of the new landslide ratio of each event versus its corresponding
accumulated rainfall of Chushui and Aiyuzih subwatershed

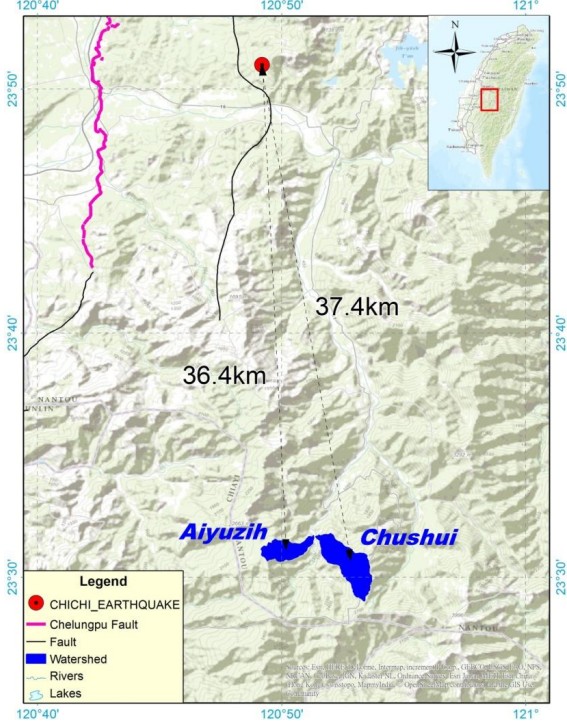

Figure18. Geographic maps of the distance from Chi-chi earthquake epicenter of Chushui and
Aiyuzih subwatershed



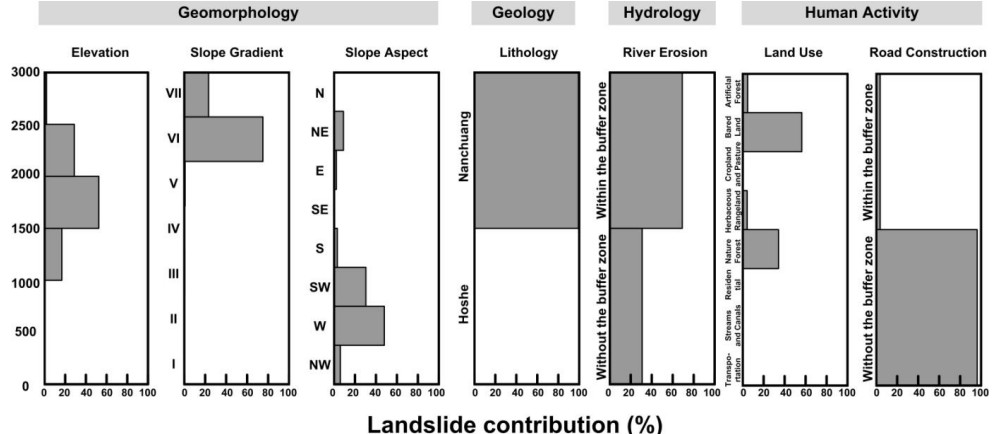

(a)  Chushui subwatersheds

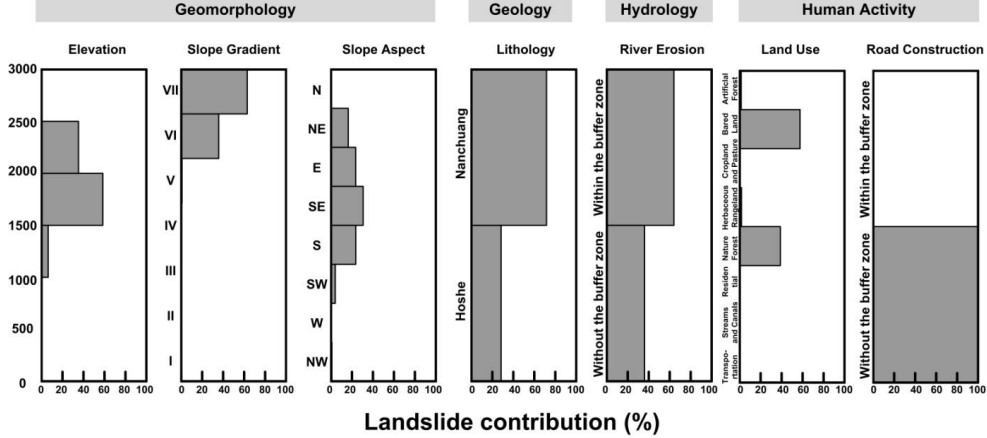

(b)  Aiyuzih subwatersheds
Figure19. Graph of the average landslide contribution of various environmental factors after
1999 Chi-Chi earthquake for Chushui and Aiyuzih subwatersheds



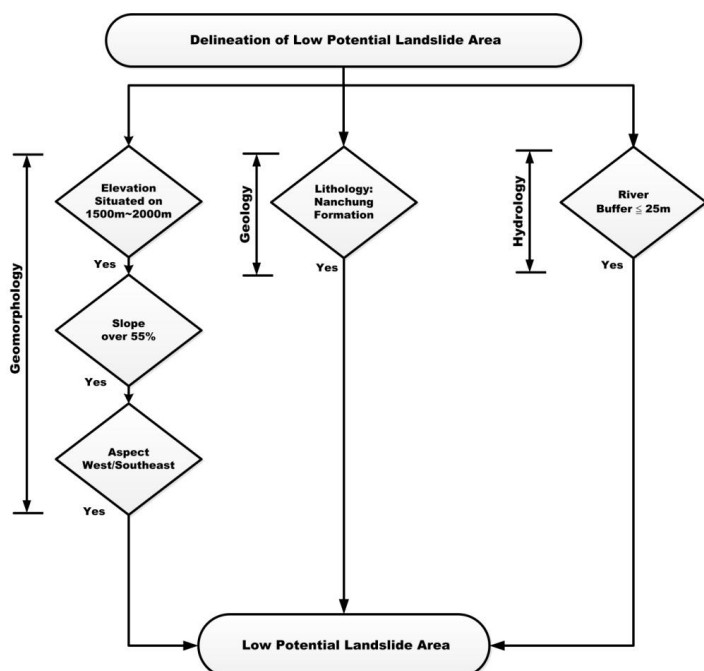

2    Figure 20. Procedure of delineation of low potential landslide area

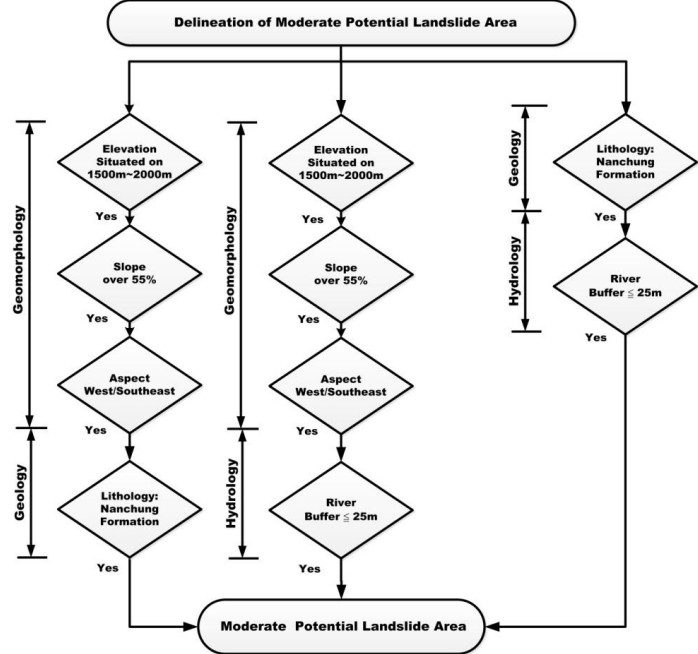

4    Figure 21. Procedure of delineation of moderate potential landslide area

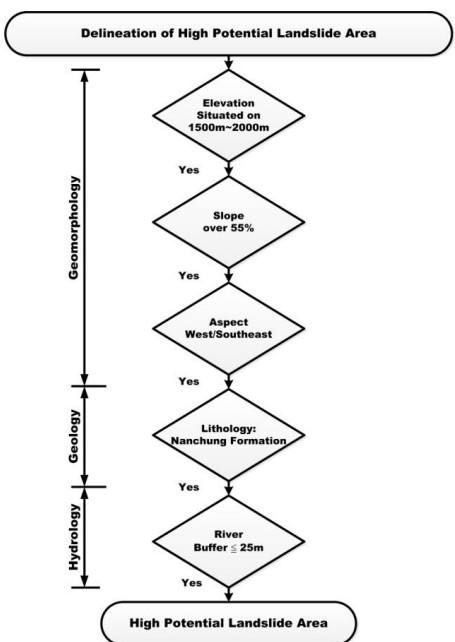

2      Figure 22. Procedure of delineation of high potential landslide area

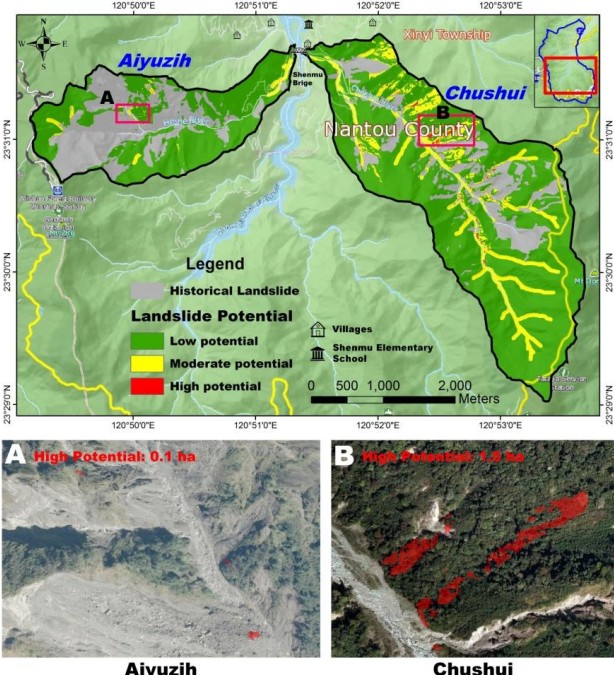

4      Figure 23. Landslide potential map of Chushui and Aiyuzih subwatersheds