# Peer review of "© Author(s) 2016. CC-BY 3.0 License."

_Natural Hazards and Earth System Sciences, 2016_

## Referee Comment (RC1) · Anonymous Referee #1 · 1 Aug 2016

The paper analyses landslide inventories maps prepared for several events in Taiwan to evaluate which are the environmental factors and cumulated rainfall that influence the generation of new landslides, including the magnification effects produced by typhoons after the 2009 Chi-Chi earthquake. The Authors analyze a large set of spot images (17 scenes) to produce landslide inventory maps. They adopted a two-step mapping method, combining a semi-automatic method based on an object-oriented approach with interpretation made by a skilled photointerpreter to adjust the classification of the polygons. The landslide inventories are analyzed considering the environmental factors and the triggering (rainfall events) adopting the Uchiughi (1971) relation. The results obtained are used to build a so-called "logical reason-based" rule set to obtain

a landslide potential map. The contents of the manuscript fit the scope of the journal. The scientific approach is not innovative, but the use of a very wide dataset is interesting and probably worth being published. Unfortunately, in my opinion, the manuscript is poorly structured and the data are not presented clearly. This make the manuscript hard to read. I recommend the Authors to rewrite completely the work. 1. Number of figures: 23 figures are too many for a single paper. The number of figures could be reduced, for example, summarizing figures from 1 to 3 into a single figure; the same applies to other cases. 2. The Introduction section is rather general and not focused on the purpose of the work. Many introductive information are found in other paragraphs such as "Spatial data and methodology" or in the firsts parts of other subparagraphs (see specific comments). 3. Hazard history of the study area. This Section is too long and not useful to make the point of the manuscript (see specific comments). 4. The whole manuscript can be made clearer; for example the "Methods and spatial data" should present strictly the data and methods used for the analysis; the Results section should present strictly the results of this study(see specific comments). 5. In this Discussion section, the Authors should discuss ALL the obtained results point by point, which is not the case in the present version of the Manuscript. The discussion of the Earth amplification Effects is not clear to me and poorly described. Moreover the subsection 5.2 and 5.3 "Combination of Causative factors" and "Landslide potential maps" should be moved in the Method and Result section, respectively. Specific comments: Page 3, lines 6-11: what do you mean by "primary contributors" and "secondary contributors"? River undercutting is a borderline factor (trigger or causative). Page 5, line 6: the analysis consider the influence of Nanchuang and Heshe formations. Please describe the two formations in detail, highlighting the existing differences and providing information about the percentage content of each formation in the two watersheds, Aiyuzih and Chushui. Moreover, the two watersheds should be described in terms of factors considered for the analysis (elevation, slope, aspect, lithology, human activities ecc.), and other irrelevant information should be removed (for example, the historical information from page 5, line 14 to page 6, line 16) or substantially reduced. Page 6,

line 17-19: irrelevant here; please remove. From page 6, line 21 to page 9: there are many portions of text which belong to the Introduction. In this Section, only the spatial data used in the analysis should appear. Page 8, line 30: "If NDVI value is less than 0.05, there is a high probability that the detected land cover/objects are landslides (see Fig.5)". This fact cannot be seen from figure 5. Moreover, the statement about the numerical value should be justified or a reference should be provided. From page 11, line 10 to page 12, line 10: please try to illustrate the artificial image identification in a clearer and synthetic way. Section 3.3: the whole section provides a lot of irrelevant information and, in my opinion, it should be substantially reduced. Page 13, line 21: "landslide area" should be "watershed area", according to Eq. (2). Page 15, lines from 2 to 10: these introduction is irrelevant to the Section, and should be removed. In the following, the Authors use the Uchihugi formula to calculate the new landslide ratio from the magnitude of the rainfall events. They modify the original formula adding a parameter, C. How do they obtain the value of C parameter quoted in figure 17? What do they mean by "initial increment landslide ratio"? Which is the physical meaning of the constant they introduce? Page 15, line 15: "However, when the rainfall parameters of Uchihugi empirical model reach the critical rainfall, the new landslide in the watershed becomes zero." This sentence is not clear to me! It seems that when the value of cumulated rainfall is larger than the value of critical rainfall the new landslide becomes zero. I checked in the article "S.-J. Chiou, et al.: Evaluating Landslides and Sediment Yields Induced by the Chi-Chi Earthquake and Heavy Rainfalls" (the suggested reference is actually only available in Japanese, which is not acceptable), in which the same formula is used and the value zero is obtained for cumulated rainfall smaller than critical rainfall, as it should be. From page 16, line 6 to page 17, line 3: the description of the temporal and spatial analysis could be used to introduce briefly paragraph 4.1 and 4.2. Please remove, or move this part to the suggested location. Page 20, lines from 2 to 8: this part fits better in the Discussion section than in the Results one; many sentences are very general ones and can be safely removed. In conclusion, I believe that, in general, the manuscript should be substantially reduced in length by

removing irrelevant information. The Discussion and Conclusions sections should be rewritten from scratch, using the results actually obtained in this work and avoiding generic comments and lengthy introductory text. Moreover, I suggest that the Authors analyze how the landslide potential map change using the combination of causative factor for the three temporal periods pre-1999 Chi-Chi earthquake, from 1999 Chi-Chi earthquake to pre- and post- typhoon Morakot. A validation of the map itself could be performed by discussing the landslide potential map obtained from each period against the observation of the next one.

---

## Referee Comment (RC2) · Anonymous Referee #2 · 6 Aug 2016

This study presents as a final result the landslide potential map of two river basins in central Taiwan, an area which is highly affected by natural hazards. The landslide inventories are prepared for 17 scenes with a connexion to several typhoon and earthquake events within a time period of 14 years. The inventories are statistically analysed including causative and triggering factors using landslide ratio and a logical empirical equation after Uchiogi (1971) to get the landslide potential map. This paper fits to the scope of the journal. The approach takes the opportunity to analyse satellite images over a longer time period to examine correlations between earthquake and rainfall events triggering landslides. The results can be seen as basis for further hazard and risk research in this area and it is worth to publish the outcomes. Nevertheless it has

to be highlighted that the manuscript is poorly structured, methods and results are not described clearly enough, input data are not presented properly. The text is hard to read. I recommend revising and adopting the entire manuscript. 1. The aim of the study should be mentioned in the abstract and the introduction not in the description of the study area (p. 6 lines 17-19) 2. There are repetitions of information on the usage of the results for further work which should be mentioned only ones. (i.e. p.6 lines26ff.) 3. There are many figures added which are not essential for readers support. Especially figure 4 is unnecessarily because it is hardly mentioned in the text. A legend is missing and does not give an additional input to the reader. 4. Please list all input data clearly with citation of the source and if available the resolution/scale for which they are suitable. 5. Try to present complete lists p.7 line 7 "GIS layers such as roads. Also describe their preparation (eg. buffer of roads) Maybe it makes sense to have an own chapter to describe data and data preparation. Used aerial photographs supporting the image identification should also be listed by date and citation. 6. On p.10 lines 22: I would like to know more about your work of classification and hierarchy and tree structure and not what should have been done. 7. Results: Landslide distribution in figure 8 there are landslide areas which are not only rising during the time. See areas post earthquake Chi-Chi (1999/10/31) total area of 18.767ha whereas pre typhoon Toraji (2011/01/20) the total landslide area is 14.465ha. How do you explain this issue?-How do you consider this circumstances in the following procedures for the landslide potential map? In figure 9 there is a gap in post-typhoon Toraji and pre typhoon Mindulle. This issues would haven been interesting to be discussed. 8. Figure 13 represents not elevation as mentioned on p.20 line 11. The figure shows slope classes. 9. Human activity: p. 23 Line 7 and after in the discussion p. 24 line 19 you mentioned that human activity causes minor or irrelevant landslide contribution. There should maybe discussed the fact, that the area is located in a very steep and montainous part of Taiwan. 10. Generally in the results and discussion chapters the final landslide potential map as the final result is mentioned very shortly. It is mentioned on p.16 line 6 "...this section utilized a dataset of complete and reliable landslide inventory maps of Shenmu area...." How

do you validate your working steps? In general validation of any of your results is required. Landslide potential maps of the different time periods are particularly suitable to evaluate and discuss the model as well as the outcomes. 11. Additionally editing remarks. The Name "Uchiogi" in the References is spelled differentially than in the text "Uchiughi". P. 18 line 21 "Fig. xx" needs a correct numerical value.

---

## Referee Comment (RC3) · Anonymous Referee #3 · 7 Aug 2016

Over all speaking, this paper did integrate techniques and data from Remote Sensing and GIS to compute quantitative indices that were in turn used to describe the changes of landslides within the study area. The use of spatial statistics to show the relationship between environmental factors and the landslides is very useful in getting insight of how landslides were triggered and can be used in a better watershed management. Especially the landslide contribution (LC) for each unique condition units (UCU) is an objective index for management purposes. The final result, Landslide Potential Map, does provide researchers and managers what and where the next step they should go to.

There are some concerns too. First of all, the spatial resolutions of SPOT5 and

FORMOSAT-2 are not identical. This might create some sort of inconsistency between classification results from different satellite. The second is about the minimum unit size of UCU. This involves the MAUP issue. The last and the most concern is the "Landslide Potential Map". This is the major contribution of this paper, low, moderate and high landslide threat classes. The authors should provide more details about how these three classes are defined

---

## Author Comment (AC1) · 24 Aug 2016

**Amendments to Proof (Reviewer #1) Date: 2016/08/24**

**Author(s) Name(s):** Kuei-Lin Fu, Bor-Shiun Lin,*, Kent Thomas, Chun-Kai Chen and Hsing-Chuan Ho

**Reference No:** nhess-2016-127

**Paper Title:** Evaluation of Environmental Factors in Landslide Prone Areas of Central Taiwan using Spatial Analysis of Landslide Inventory Maps

| Item No. | Original Paper | | | Comments | Author's Response |
|---|---|---|---|---|---|
| | Page | Line | Text | | |
| 1 | 37-54 | - | - | Number of figures: 23 figures are too many for a single paper. The number of figures could be reduced, for example, summarizing figures from 1 to 3 into a single figure; the same applies to other cases. | Thanks for the comments. To avoid excessive figures and reduce unnecessary information, the authors will try the best to combine those similar GIS-data layers, or statistical results of identical regions into single figures within reason as to not diminish the quality of the study and then polish the revised paper to make it more readable, the data more clear. |
| 2 | 3 | - | - | The Introduction section is rather general and not focused on the purpose of the work. Many introductive information are found in other paragraphs such as "Spatial data and methodology" or in the firsts parts of other subparagraphs (see specific comments) | Thanks for the comments. Under your direct indications, the Introduction section of the manuscripts will be rewritten substantially and introductory information found in other sections will be moved to this section. After revision, the full paper will be expected to be well-structured, more concise and highlighted the research results and data value. |
| 3 | 4-6 | - | - | Hazard history of the study area. This Section is too long and not useful to make the point of the manuscript (see specific comments). | Thanks for the comments. Shenmu area has been affected by serious sediment-related disasters and the hazard history information is used to point out and classify the landslide areas of differing proneness. The section aims to deliver hazard history of over 20 years and the effects of heavy rainfall and typhoon events highlighting and strengthening the legitimacy of the research and the context which it represents. Without the hazard history, the validity of the report becomes more questionable and is seemingly incomplete. The inclusion of hazard history information in landslide research is seen as industry-standard as can be seen by many different researches. Nonetheless, the authors will |

| Item No. | Original Paper | | | Comments | Author's Response |
|---|---|---|---|---|---|
| | Page | Line | Text | | |
| | | | | | simplify the whole section content and reduce the length of each paragraph by removing irrelevant or information not useful for revealing or improving upon the above points. |
| 4 | 6-8 16-23 | | - | The whole manuscript can be made clearer; for example the "Methods and spatial data" should present strictly the data and methods used for the analysis; the Results section should present strictly the results of this study(see specific comments) | **Thanks for the comments.** The two sections described above "Methods and spatial data" and "Results" will be reconstructed and rephrased to reflect data and methods and results of this study. |
| 5 | | | | In this Discussion section, the Authors should discuss all the obtained results point by point, which is not the case in the present version of the Manuscript. The discussion of the Earth amplification Effects is not clear to me and poorly described. Moreover the subsection 5.2 and 5.3 "Combination of Causative factors" and "Landslide potential maps" should be moved in the Method and Result section, respectively. | **Thanks for the comments.** The obtained results have been described point by point in the Results section. Earthquake Amplification Effect is not the main contribution of this paper and has been discussed by previous researchers mentioned in references, therefore it is unnecessary for a lengthy discussion on this topic. The authors will only keep "Combination of Causative Factors" and "Landslide Potential" to the sections suggested enhancing the value and readability of the study. The observation related to Earthquake Amplification Effect are shortly concluded and removed into the Conclusion Section. |
| 6 | P3 | 6-11 | - | what do you mean by "primary contributors" and "secondary contributors"? River undercutting is a borderline factor (trigger or causative) | **Thanks for the comments.** In this study, we refer to some researcher papers which discuss landslide occurrence and utilize some key terms associated with landslide occurrence. One such term is the triggering factor and another is causative factor. The triggering factor of landslides occurrence is associated with dynamic characteristics which means landslides occur when unstable rock and soil masses on slopes are disturbed by agents of natural or human activities such as heavy rainfall, typhoons, earthquakes, river undercutting or road construction. The causative factor of landslides occurrence is associated with |

| Item No. | Original Paper | | | Comments | Author's Response |
|---|---|---|---|---|---|
| | Page | Line | Text | | |
| | | | | | static characteristics with spatial characteristics with a given environmental that inherit itself the potential causes of vulnerability such as topography, geology, land use. |
| | | | | | To avoid misunderstandings, the authors will remove the words above "primary contributors" and "secondary contributors". The original paragraphs will be amended in detail and also add some reference to support the above description. |
| 7 | P5 | 6 | | Page 5, line6: the analysis consider the influence of Nanchuang and Heshe formations. Please describe the two formations in detail, highlighting the existing differences and providing information about the percentage content of each formation in the two watersheds, Aiyuzih and Chushui. Moreover, the two watersheds should be described in terms of factors considered for the analysis (elevation, slope, aspect, lithology, human activities ecc.), and other irrelevant information should be removed (for example, the historical information from page 5, line 14 to page 6, line 16) or substantially reduced. | Thanks for the comments. The Shenmu area is crossed by three primary geologic structures: the northeast-southwest Heshe Anticline and Tungfu Syncline and the Chen-yo-lan River Fault. These mountain slopes are covered with dense forests and were built up by the Nanchuang and Heshe formation. These formations consists mainly of hard, dark grey argillite and grey slate with thinly bedded muddy sandstone, which are prone to severe weathering and become weak layers in the rock strata (see the Figure). Its percentage content of each formation in the Aiyuzih (45.36% of Heshe formation and 54.64% of Nanchuang formation) and Chushui watersheds (0.61% of Heshe formation and 99.39% of Nanchuang formation) will be provided into revised paper. In addition, environmental factors affecting landslide occurrence considered for the analysis (elevation, slope, aspect, lithology, human activities ecc.) will first be substantially reduced and then introduced in the heading of manuscript. |

| Item No. | Original Paper | | | Comments | Author's Response |
|---|---|---|---|---|---|
| | Page | Line | Text | | |
| | | | | |  |
| | | | | | **Figure. Photos of dark grey argillite and grey slate** |
| 8 | P6 | 17-19 | - | **Page 6, line 17-19: irrelevant here; please remove.** | **Thanks for the comments.** |
| | | | | **From page 6, line 21 to page 9: there are many portions of text which belong to the Introduction. In this Section, only the spatial data used in the analysis should appear.** | **Irrelevant information in Page 6, line 17-19: would be reduced or deleted. And, introductory information found in page 6, line 21 to page 9 would be removed to the Introduction** |
| 9 | P10 | 30 | - | **Page 10, line 30: "If NDVI value is less than 0.05, there is a high probability that the detected land cover/objects are landslides (see Fig.5)". This fact cannot be seen from figure 5. Moreover, the statement about the numerical value should be justified or a reference should be provided.** | **Thanks for the comments.** |
| | | | | | **The original sentence has not described NDVI method or value precisely so this causes misunderstandings.** |
| | | | | | **In a single, identical area, satellite imagery acquirement may be affected by seasonal changes, the angle of incidence of the Sun, atmospheric radiation and capture angles, which results in inconsistence of NDVI range for each difference period images.** |
| | | | | | **Therefore, this paper chose some of samples plot on the single** |

| Item No. | Original Paper | | | Comments | Author's Response |
|---|---|---|---|---|---|
| | Page | Line | Text | | |
| | | | | | image to define NDVI threshold to separate vegetation and non-vegetation areas with reference GIS-layers such as roads and land use maps to identify whether it belongs to landslide area.

The revised manuscript will, for example, take given satellite images and utilize NDVI results to illustrate vegetation change and the original sentence will also be amended according to the above mentioned. Additionally to increase clarity, some references to NDVI method of land cover detection will be added. |

| Item No. | Original Paper | | | Comments | Author's Response |
|---|---|---|---|---|---|
| | Page | Line | Text | | |
| | | | | |
[Figure]
 |

| Item No. | Original Paper | | | Comments | Author's Response |
|---|---|---|---|---|---|
| | Page | Line | Text | | |
| 10 | | - | - | From page 11, line 10 to page 12, line 10: please try to illustrate the artificial image identification in a clearer and synthetic way. | Thanks for the comments.

The authors will add the graph to clearly demonstrate artificial image identification. |
| 11 | 12-13 | | | Section 3.3: the whole section provides a lot of irrelevant information and, in my opinion, it should be substantially reduced. | Thanks for the comments.

The authors will follow your suggestion to reduce some information to highlight the standpoint of Section 3.3. |
| 12 | 13 | 21 | | Page 13, line 21: "landslide area" should be "watershed area", according to Eq. (2). | Thanks for the comments.

The paragraph is amend as below,

*Similarly, the new landslide ratio as the ratio of new landslide area to the watershed area.* |
| 13 | | | | Page 15, lines from 2 to 10: these introduction is irrelevant to the Section, and should be removed.

In the following, the Authors use the Uchihugi formula to calculate the new landslide ratio from the magnitude of the rainfall events. They modify the original formula adding a parameter, C. How do they obtain the value of C parameter quoted in figure 17? What do they mean by "initial increment landslide ratio"? Which is the physical meaning of the constant they introduce?

Page 15, line 15: "However, when the rainfall parameters of Uchihugi empirical model reach the critical rainfall, the new landslide in the watershed becomes zero." This sentence is not clear to me! It seems that when the value of cumulated rainfall is larger than the value of critical rainfall the new landslide becomes zero. I checked in the article "S.-J. Chiou, et al.: Evaluating Landslides and Sediment Yields Induced by the Chi-Chi Earthquake and Heavy Rainfalls" (the suggested reference is actually only | Thanks for the comments.

In some parts of the paper Uchiogi is mispelled as Uchiughi, these errors will be corrected in the revised paper.

According to Uchiogi formula, one can assume that for a given watershed, triggered landslide emperica equation for a rainfall event under 200mm of critical rainfall can be stated as follows:

$NLR(\%) = \dfrac{ILA}{WSA} \approx K \times 10^{-6}(R_A - 200)^2 \ R_A \geq 200$

One can find that if the accumulated rainfall is up to 200mm and is inputted into this formula, the estimate of the new landslide ratio by this event can be calculated as follows:

$NLR(\%) = \dfrac{ILA}{WSA} \approx K \times 10^{-6}(200 - 200)^2 = 0$

The calculated result shows that the new landslide ratio is zero which means no landslides or slope failures occurred during |

| Item No. | Original Paper | | | Comments | Author's Response |
|---|---|---|---|---|---|
| | Page | Line | Text | | |
| | | | | **available in Japanese, which is not acceptable), in which the same formula is used and the value zero is obtained for cumulated rainfall smaller than critical rainfall, as it should be.** | **this event which does not match with physical phenomena of the exceedance of critical rainfall for slope failure.**

**To modify the imperfection of the original formula, the authors suggest the addition of one constant(C) to the original formula. C aims to represent that when the accumulated rainfall equals or exceeds the critical rainfall a certain amount of new landslide occurs in a watershed. The correction will make the original formula more reasonable and enhances its applicability. The modified formula is as follows:**

$$NLR(\%) = \frac{ILA}{WSA} \approx C + K \times 10^{-6}(R_A - 200)^2 \quad R_A \geq 200$$

**In terms of obtaining the value of C parameter, the data for each rainfall-induced landlide event and its corresponding accumulated rainfall will be compiled for statistical regression based analysis on the suggesting governing formula (see the figure). The figure shows that the C parameter is a constant and seems like a truncated value of the y-axis(new landslide ratio).**

**From the perspective of physical significance, it implies when a rainfall event reaches its critical rainfall, this parameter(C** |

| Item No. | Original Paper | | | Comments | Author's Response |
|---|---|---|---|---|---|
| | Page | Line | Text | | |
| | | | | | constant) represents the minimum amount of new landslide area in a watershed. |
| | | | | | In view of these points, the authors will arrange the above mentioned descriptions and revise the original sentence to make the manuscript more readable and clear. |
| 14 | 16 | 6 | | From page 16, line 6 to page 17, line 3: the description of the temporal and spatial analysis could be used to introduce briefly paragraph 4.1 and 4.2. Please remove, or move this part to the suggested location. | Thanks for the comments.

The authors will remove those sentence such in from page 16, line 6 to page 17, line 3 to the heading of Subsection 4.1 and 4.2 respectively under your suggestions. |
| 15 | 20 | 2-8 | | Page 20, lines from 2 to 8: this part fits better in the Discussion section than in the Results one; many sentences are very general ones and can be safely removed. | Thanks for the comments.

The authors will follow your suggestion to remove sentences in Page 20, lines from 2 to 8 into the Discussion section. And, if the sentences are very general ones, they would be safely removed. |
| 16 | | | | In conclusion, I believe that, in general, the manuscript should be substantially reduced in length by removing irrelevant information.

The Discussion and Conclusions sections should be rewritten from scratch, using the results actually obtained in this work and avoiding generic comments and lengthy introductory text.

Moreover, I suggest that the Authors analyze how the landslide potential map change using the combination of causative factor for the three temporal periods pre-1999 Chi-Chi earthquake, from 1999 Chi-Chi earthquake to pre- and post- typhoon Morakot. A validation of the map itself could be performed by discussing the landslide potential map obtained from each period against the observation of the next one. | Thanks for the comments.

The author will follow your suggestion to remove relevant information and reconstruct the Discussion and Conclusions sections based on main obtained results. In addition, landslide potential map of the three temporal periods pre-1999 Chi-Chi earthquake, from 1999 Chi-Chi earthquake to pre- and post-typhoon Morakot would be also discussed and validated in the revised manuscript. |

---

## Author Comment (AC2) · 24 Aug 2016

**A m e n d m e n t s   t o   P r o o f   ( R e v i e w e r   # 2 )   D a t e : 2 0 1 6 / 0 8 / 2 4**

**Author(s) Name(s):** Kuei-Lin Fu, Bor-Shiun Lin,*, Kent Thomas, Chun-Kai Chen and Hsing-Chuan Ho

**Reference No:** nhess-2016-127

**Paper Title:** Evaluation of Environmental Factors in Landslide Prone Areas of Central Taiwan using Spatial Analysis of Landslide Inventory Maps

| Item No. | Original Paper | | | Comments | Author's Response |
|---|---|---|---|---|---|
| | Page | Line | Text | | |
| 1 | 6 | 17-19 | - | The aim of the study should be mentioned in the abstract and the introduction not in the description of the study area (p. 6 lines 17-19) | Thanks for the comments. The authors will remove the aim of the study from the study area and properly highlight it in the introduction and abstract. |
| 2 | 6
7 | 26-31
1-17 | | There are repetitions of information on the usage of the results for further work which should be mentioned only ones. (i.e. p.6 lines26ff.) | Thanks for the comments. Information on the usage of the results would be reduced to only represent the standpoint of the Section 3. |
| 3 | 40 | | | There are many figures added which are not essential for readers support. Especially figure 4 is unnecessarily because it is hardly mentioned in the text. A legend is missing and does not give an additional input to the reader | Thanks for the comments. Figure 4 is a sketch image rendering map that illustrates a source of satellite imagery acquirement. The author will add legend to figure 4 and also introduce the importance of satellite imagery acquirement related an event to increase the readability of manuscript. |
| 4 | | | | Please list all input data clearly with citation of the source and if available the resolution/scale for which they are suitable. | Thanks for the comments. The author will add a table to list all of used data and its information including time, citation of the source, resolution/scale and the application. |
| 5 | 10 | 7 | | Try to present complete lists p.10 line 7 "GIS layers such as roads. Also describe their preparation (eg. buffer of roads) Maybe it makes sense to have an own chapter to describe data and data preparation. Used aerial photographs supporting the image identification should also be listed by date and citation. | Thanks for the comments. The authors will add a graph to present the used GIS layers and describe the step "Importation of Satellite images and references" in Subsection 3.2.1. Also, all of the used GIS layers and aerial photographs supporting the image identification will also be listed by date and citation. |

| Item No. | Original Paper | | | Comments | Author's Response |
|---|---|---|---|---|---|
| | Page | Line | Text | | |
| | | | | |  |
| 6 | 10 | 22 | | On p.10 lines 22: I would like to know more about your work of classification and hierarchy and tree structure and not what should have been done. | Thanks for the comments. The classification and hierarchy and tree structure for landslide detection by interface of the used automated program will be illustrated into graph as schematic layout in the revised manuscript |
| 7 | | | | Results: Landslide distribution in figure 8 there are landslide areas which are not only rising during the time. See areas post earthquake Chi-Chi (1999/10/31) total area of 18.767ha whereas pre typhoon Toraji (2011/01/20) the total landslide area is 14.465ha. How do you explain this issue? In figure 9 there is a gap in post-typhoon Toraji and pre typhoon Mindulle. This issues would have been interesting to be discussed. How do you consider this circumstances in the following procedures for the landslide potential map? | Thanks for the comments. This study establishes the event-based landslide inventory using multi-stage satellite imagery interpretation for Chushui and Aiyuzih subwatersheds in the Shenmu area which only involved ten extreme meteorological events and one extreme earthquake event. The period of post earthquake Chi-Chi (1999/10/31) to pre typhoon Toraji (2011/01/20) have not suffered high prolonged rainfall event so the decreased landslide areas can be attributed to the effects of environmental natural vegetation recovery which increases a slopes ability to mitigate |

| Item No. | Original Paper | | | Comments | Author's Response |
|---|---|---|---|---|---|
| | Page | Line | Text | | |
| | | | | | **landslides or from a geotechnical standpoint improves the factor of safety of the slope.** |
| | | | | | **Similarly, post-typhoon Toraji and pre typhoon Mindulle has no significant rainfall events so that the decreased areas can be attributed to the effects of environmental natural vegetation recovery. Secondly, the procedures for the landslide potential map in our manuscript is aimed to present and delineate the future landslide for areas which have well-vegetated land cover and presently have no evidence of landslide activity.** |
| | | | | | **Accordingly, the authors will rewrite these sentences to make it clear.** |
| 8 | 20 | 11 | | **Figure 13 represents not elevation as mentioned on p.20 line 11. The figure shows slope classes.** | **Thanks for the comments.** **The figure number will be corrected as "12".** |
| 9 | | | | **Human activity: p.23 Line 7 and after in the discussion p. 24 line 19 you mentioned that human activity causes minor or irrelevant landslide contribution. There should maybe discussed the fact, that the area is located in a very steep and mountainous part of Taiwan** | **Thanks for the comments.** **The area where the majority of landslides are found in Shenmu is generally a very steep and mountainous part of Taiwan where human activity is minimal and causes irrelevant to minor contribution to the total landslide area of Shenmu. The author will add the sentence to explain the above.** |
| 10 | | | | **Generally in the results and discussion chapters the final landslide potential map as the final result is mentioned very shortly.** **It is mentioned on p.16 line 6 ": : :this section utilized a dataset of complete and reliable landslide inventory maps of Shenmu area: : :." How do you validate your working steps? In general validation of any of your results is required. Landslide potential maps of the different time periods are particularly suitable to evaluate and discuss the model as well as the outcomes.** | **Thanks for the comments.** **The author will follow your suggestion to remove relevant information and reconstruct the Discussion and Conclusions sections based on main obtained results. In addition, landslide potential map of the three temporal periods pre-1999 Chi-Chi earthquake, from 1999 Chi-Chi earthquake to pre- and post-typhoon Morakot would be also discussed and validated in the revised manuscript.** |

| Item No. | Original Paper | | | Comments | Author's Response |
|---|---|---|---|---|---|
| | Page | Line | Text | | |
| 11 | 18 | 21 | | Additionally editing remarks. The Name "Uchiogi" in the References is spelled differentially than in the text "Uchiughi". P. 18 line 21 "Fig. xx" needs a correct numerical value | Thanks for the comments.

That is a misspelled word. The text "Uchiughi" of the manuscript should be amended as "Uchiogi" according to the references. And, P. 18 line 21 Fig. number is "11". |

---

## Author Comment (AC3) · 24 Aug 2016

**A m e n d m e n t s   t o   P r o o f   ( R e v i e w e r   # 3 )   D a t e : 2 0 1 6 / 0 8 / 2 4**

**Author(s) Name(s):** Kuei-Lin Fu, Bor-Shiun Lin,\*, Kent Thomas, Chun-Kai Chen and Hsing-Chuan Ho

**Reference No:** nhess-2016-127

**Paper Title:** Evaluation of Environmental Factors in Landslide Prone Areas of Central Taiwan using Spatial Analysis of Landslide Inventory Maps

| Item No. | Original Paper | | | Comments | Author's Response |
|---|---|---|---|---|---|
| | Page | Line | Text | | |
| 1 | | | - | the spatial resolutions of SPOT5 and FORMOSAT-2 are not identical. This might create some sort of inconsistency between classification results from different satellite. | Thanks for the comments. This study select satellite imagery sources that its acquisition date should be close to disaster events for reflecting real geomorphological change. So, all of the satellite imagery sources could not be identical or provided by single satellite data acquisition system. Of course, SPOT series and FORMOSAT-2 satellites had different spatial resolutions. To deal with the problem, this study collects and uses aerial photo close to the event-based landslide inventory to which systematically improves the quality of data content and data structure so that it can definitely promote accuracy and reduce inconsistency between classified results from different satellites |
| 2 | | | | The second is about the minimum unit size of UCU. This involves the MAUP issue. | Thanks for the comments. This study resamples source data layers to the same resolution to reduce zone effect during executing spatial analysis to reduce the effects of UCU involved MAUP issue. |
| 3 | | | | The last and the most concern is the "Landslide Potential Map". This is the major contribution of this paper, low, moderate and high landslide threat classes. The authors should provide more details about how these three classes are defined | Thanks for the comments. Regarding to the principle of classification of "Landslide Potential Map, one could use the concept of Venn Diagrams to explain the definition of low, moderate and high landslide threat classes. A Venn diagram (also called a set diagram or logic diagram) is a diagram that shows all possible logical relations between finite collections of different sets which can illustrates simple set relationships in probability. In this study, |

| Item No. | Original Paper | | | Comments | Author's Response |
|---|---|---|---|---|---|
| | Page | Line | Text | | |
| | | | | | there are the three sets of environmental factors which is a combination of geomorphology, geology, hydrology to dominate landslide contribution within the study area. A Venn diagram of these three sets in S region (landslide potential space)could be illustrated as below |

In this figure, there are seven relationships among these three set in landslide potential space. Based on the Venn diagram, one can calculate the landslide probability of a given event to qualitatively represent the landslide potential as low, moderate and high. Namely, landslide probability is getting higher which the landslide area has higher potential.

Case 1 :Only one set without intersections (Event A) will trigger the landslide probability as calculated

The probability of Case 1 is equal to 3/7

Case 2 : Intersections of each two sets (Event B) including Event A will trigger the landslide probability as

| Item No. | Original Paper | | | Comments | Author's Response |
|---|---|---|---|---|---|
| | Page | Line | Text | | |
| | | | | | **calculated** |
| | | | | | **The probability of Case 2 is equal to 6/7(=3/7+3/7)** |
| | | | | | **Case 3: Intersections of three sets (Event C) including Event A and Event B will trigger the landslide probability as calculated** |
| | | | | | **The probability of Case 3 is equal to 7/7(=3/7+3/7+1/7)** |
| | | | | | **According to the calculated results of landslide probability, Case 1, Case 2 and Case 3 will be defined as low, moderate and high landslide potential classes.** |
| | | | | | **For avoiding misunderstanding, the authors will rewrite and add more detailed definition about the principle of classification of "Landslide Potential Map" in the revised manuscript.** |